# VIBeID: A Structural Vibration-based Soft Biometric Dataset for Human Gait Recognition

## Abstract

We present VIBeID, a dataset and benchmark designed for advancing non-invasive human gait recognition using structural vibration. Structural vibrations, produced by the rhythmic impact of the toe and heel on the ground, are distinct and can be used as a privacy-preserving and non-cooperative soft-biometric modality. We curated the largest dataset VIBeID consists of footfall generated structural vibrations of 100 subjects. Existing datasets in this field typically include around ten subjects and lack comprehensive exploration of domain adaptation. To thoroughly explore the domain adaptation aspect of this biometric approach, we recorded vibration data on three distinct floor types (wooden, carpet, and cement) and at three distances from the geophone sensor (1.5 m, 2.5 m, and 4.0 m), involving 40 and 30 subjects, respectively. Additionally, we benchmarked our dataset against video recordings from 15 individuals in an outdoor setting. Beyond providing 88 hours of raw vibration data, VIBeID establishes a comprehensive benchmark for a) person identification: where the aim is to recognize individuals through their unique structural vibrations, b) domain adaptation: assessing model performance across different walking surfaces and sensor positions, and c) multi-modal comparison: comparing vibration-based and vision-based identification methods. Our experiments, using both machine learning and deep learning approaches, establish a baseline for future research in this field, and introduce a large-scale dataset for the broader machine learning community. [1]

## 1 Introduction

Structural vibration-based person identification is an emerging topic in the field of soft biometrics (Pan et al., 2017; Anchal et al., 2020). As humans walk, our bodies exert an impact force on the ground, generating vibrations that propagate through the structure. These structural vibrations are unique to each individual as they depend on various factors such as height, weight, gait strides, stride length along with structural properties, and background noises (Succi et al., 2000). Our technique uses these vibrations to identify individuals. This paper introduces VIBeID, a novel dataset containing structural vibration signals recorded by geophone sensors from human walking. VIBeID is designed to address three key questions: a) Is it possible to accurately identify individuals from a large population based on their unique structural vibration patterns?, b) Can humans be identified at varying distances from the sensor and on different floor types using the same model?, and c) How does the performance compare to established modalities like camera?

Early biometric research was primarily focused on physiological traits, also known as "hard biometrics", such as fingerprints, iris scans, and facial recognition (Jain et al., 2007; Dantcheva et al., 2015). While "hard biometrics" offer uniqueness, they can be inflexible in deployment scenarios, where non-intrusive or remote identification methods performs better (Zewail et al., 2004). "Soft biometrics" like gait, voice, and structural vibration, prioritizes convenience over uniqueness, have gained traction to address this limitation. (Jain et al., 2004), has defined the ideal characteristics for "soft biometrics" as, a) inexpensive computation, b) usability from a distance, and c) functionality with uncooperative subjects. Our proposed system aligns perfectly with these criterias. Structural vibrations offer many advantages over existing methods, such as being low-cost, non-invasive, less

---

[1]*The project page, along with the dataset and code, is available at `https://vibeidiclr.github.io/`, under CC BY-NC-SA 4.0 License.

computationally complex, and free of privacy issues. Prior studies has established the potential of using structural vibration as behavioral biometrics (Pan et al., 2017; Anchal et al., 2020). However, such works relied on limited datasets, impeding the development of robust identification methods. Our research fills a crucial gap by compiling the first comprehensive dataset of 100 individuals, including task-specific sub-datasets totaling 88.66 hours of structural vibration signals recorded with a single geophone. We used standard models such as ResNet-18 and ResNet-50 for benchmarking. Our baseline results show the possibility of identifying humans across different underlying structures and at distinct distances from the sensor. Additionally, we validated our approach using vibration data alongside camera recordings for outdoor environment.

Our key contributions can be summarised as :

- **Dataset Development**: We introduce a novel dataset repository of structural vibration signals categorized as follows:

    - Person Identification: Raw vibration data collected from 100 individuals walking on a single floor within a 50-100 square meter area.
    - Multi-Distance: Data from 30 subjects walking at three pre-defined distances (1.5 m, 2.5 m, and 4.0 m) from a geophone sensor.
    - Multi-Structure: Data from 40 subjects walking on three distinct floor surfaces (wood, carpet, and cement).
    - Multi-Modal Comparison: Data from 15 subjects collected in an outdoor environment using two cameras and a geophone sensor.

- **Benchmark Establishment**: We demonstrate the utility of the dataset by performing analyses on three distinct use cases:

    - Multi-Class Classification: This analysis employs both traditional machine learning and deep learning algorithms for person identification.
    - Domain Adaptation: We investigate the effectiveness of domain adaptation techniques for addressing potential variations in real-world deployments.
    - Multi-Modal Comparison: Performance comparison of vision-based and vibration-based identification methods using data from both the camera and the geophone sensor.

- **Experimental Evaluation**: We conduct a comprehensive evaluation of existing research methods on the dataset to explore their effectiveness for various tasks, where applicable.

## 2 RELATED WORK

This section delves into various soft biometric modalities, with a specific focus on person identification through human movement (gait). We categorize existing modalities as follows:

### 2.1 RELATED WORK ON PERSON IDENTIFICATION USING STRUCTURAL VIBRATION

As shown in Table 1, early research focused on establishing proof-of-concept using small datasets, typically involving around ten participants (Pan et al., 2017; Anchal et al., 2020; Dong & Noh, 2023; Mirshekari et al., 2018; Chakraborty & Kar, 2023; Xu et al., 2024). These studies aimed to demonstrate that unique characteristics of an individual's walking pattern can be recorded and classified through the vibrations transmitted to the supporting structure. However, as highlighted by (Mirshekari et al., 2018), data scarcity is a serious bottleneck, where extensive experiments with humans have not been conducted. Another aspect of these early investigations is the emphasis on mathematical modeling of the vibration signals, which prioritized developing models that capture the physical dynamics of the system rather than focusing on robust statistical analysis of the acquired data (Succi et al., 2000; Mirshekari et al., 2018). Additionally, previous datasets were highly supervised and tailored to singular use cases (training and testing models on data from the same underlying structure), limiting their applicability to real-world scenarios. We adopted a comprehensive approach to address this limitation by creating a large dataset for person identification and task-specific datasets. This strategy enhances the real-life applicability of our research.

Table 1: Overview of soft biometric datasets for person identification using various sensor modalities

| Author(s) | Sensor Modality | Dataset Description | | | Environment |
| --- | --- | --- | --- | --- | --- |
| | | Person(s) | Total Samples† | Domain | |
| (Yu et al., 2006) | Camera | 20 | 240 | 1 | Outdoor |
| (Zheng et al., 2011) | Camera | 120 | 13,640 | 1 | Indoor |
| (Tan et al., 2006) | Camera | 153 | 1530 | 1 | Outdoor |
| (Song et al., 2022) | Camera | 1014 | 778,752 | 3 | Outdoor |
| (Burdack et al., 2020) | Pressure Plates | 42 | 90 | 1 | Indoor |
| (Derlatka & Parfieniuk, 2023) | Pressure Plates | 324 | 324 | 1 | Indoor |
| (Ngo et al., 2014) | IMU | 744 | 37,500 | 1 | Indoor |
| (Shen et al., 2023) | Event cameras | 20 | 4000 | 1 | Indoor |
| (Wang et al., 2022) | LIDAR | 1050 | 25,239 | 1 | Outdoor |
| (Pan et al., 2017) | Geophone | 10 | - | 1 | Indoor |
| (Anchal et al., 2020) | Geophone | 10 | 7750 | 1 | Indoor |
| (Xu et al., 2024) | Geophone | 10 | 12,278 | 4 | Indoor |
| VIBeID (Ours) | Geophone | 100 | 144,371 | 1 | Indoor and Outdoor |
| | | 30‡ | 75,675 | 3 | |
| | | 40* | 142,526 | 3 | |
| | | 15 | 2,171 | 1 | |

† Samples (for sensor data) is equivalent to Sequence (for camera data)
‡Distances: VIBeID A2.1 (1.5 m),VIBeID A2.2 (2.5 m) and VIBeID A2.3 (4.0 m) annular from seismic sensor
∗3 structures viz, wood (VIBeID A3.1), carpet (VIBeID A3.2),and concrete (VIBeID A3.3)

## 2.2 RELATED WORK ON PERSON IDENTIFICATION OTHER THAN STRUCTURAL VIBRATION

Human movement (gait), the unique way individuals walk, is a valuable biometric for identification using cameras, particularly over long distances. Significant research has been conducted to capture and analyze human gait patterns. Previous studies have concentrated on various features extracted from walking videos, such as Gait Energy Image (Han & Bhanu, 2005), Gait Entropy Image (Bashir et al., 2009), and Gait Flow Image (Lam et al., 2011). As shown in Table 1, early research was conducted using limited datasets of twenty people (Yu et al., 2006). Recent work, however, has shifted towards large-scale datasets, incorporating both indoor and outdoor walking scenarios from various perspectives (Song et al., 2022). Our dataset aims to pave the way for further exploration in structural vibration analysis, similar to the advancements seen in video-based research. Despite the advantages of traditional gait-based video surveillance systems, it face limitations due to low-light conditions, restricted line-of-sight, adverse weather conditions, and extensive data processing (Song et al., 2022). A notable drawback of vision-based systems is the privacy concerns and user discomfort associated with continuous monitoring. Structural vibration analysis offers a promising alternative that addresses these issues. LiDAR offers a strong alternative to RGB cameras (Shen et al., 2023), but its high cost and need for a clear line of sight pose challenges, especially indoors. Event stream-based gait recognition (Wang et al., 2022) can produce sparse, noisy data in low-contrast or variable lighting conditions.

As shown in Table 1, Pressure plates and Inertial Measurement Units (IMUs) are prominent soft biometric modalities based on human movement (Ngo et al., 2014; Burdack et al., 2020; Derlatka & Parfieniuk, 2023). Pressure plates capture the unique Ground Reaction Forces (GRFs) exerted by the foot during walking by being placed directly beneath it. Pressure plates exhibit limitations as a soft biometric modality due to their requirement for direct user interaction. Deployment often requires explicit user consent, hindering their applicability in scenarios demanding unobtrusive identification (Leporace et al., 2015). IMUs, often embedded in wearable devices, offer gait-based identification through their ability to track body movements (Ngo et al., 2014). IMUs attachment to the body can be perceived as intrusive (Subramanian & Sarkar, 2018). In contrast, structural vibration-based monitoring is non-intrusive and does not require direct body contact.

## 3 BUILDING VIBeID DATASET

### 3.1 WHAT IS FOOTSTEP INDUCED STRUCTURAL VIBRATION?

During walking, our body apply forces to the ground or platform, which help us move forward and keep us balanced. These forces propagate through the structure as horizontal and vertical waves

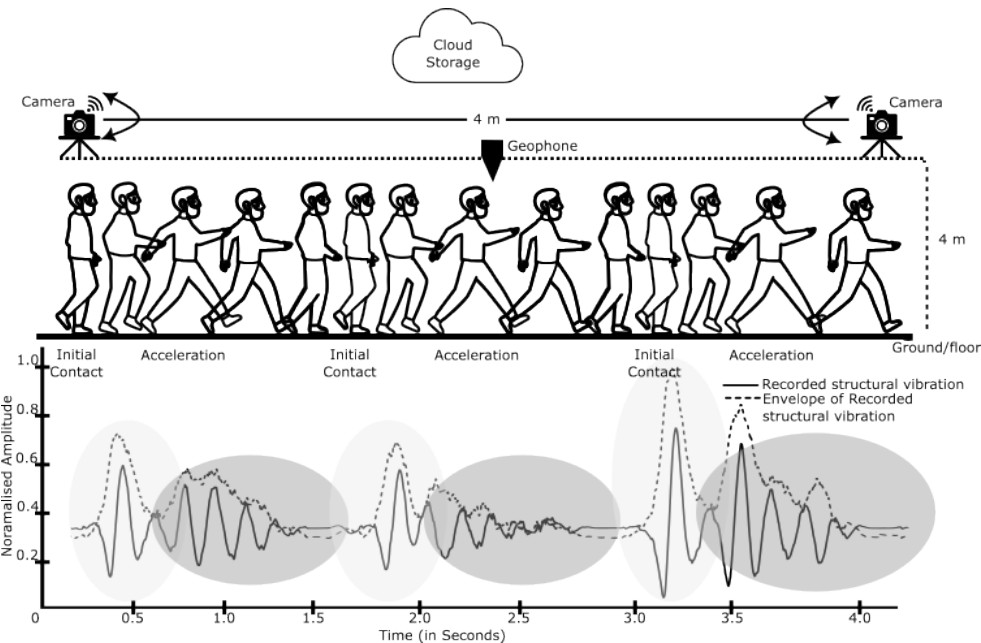

Figure 1: Visual depiction of VIBeID's data collection framework showcasing structural vibration signals, and signal envelope (Hilbert Transform) in outdoor environment

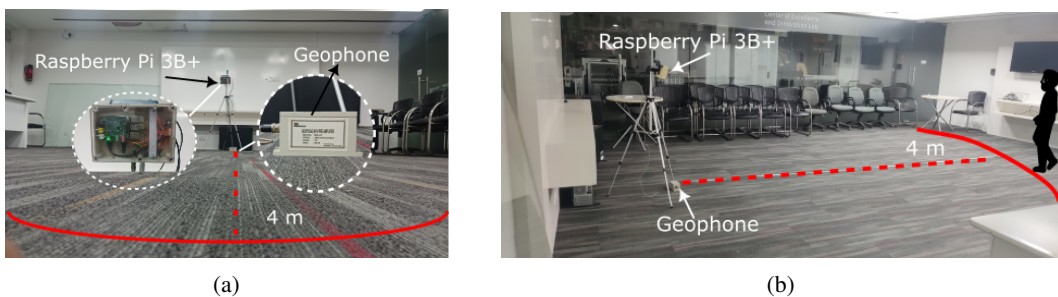

Figure 2: Data collection Setup of VIBeID A3.2 (a) the front view of setup, and (b) the side view

or vibrations (Succi et al., 2000). Our work focuses on the vertical vibrations due to their unique individual characteristics (Pan et al., 2017; Dong & Noh, 2023). As shown in Figure 1, foot strikes (initial contact) generate higher frequency vibrations than lift-off (acceleration) due to the impulsive force during the strike. We extract the envelope of the signal, using Hilbert transform, to visualise the overall variation by subsiding the rapid oscillations (Cohen, 1995; Anchal et al., 2020). This reveals a distinct pattern indicating cyclical changes in the signal corresponding to the phases of contact and acceleration. Early research has tried to model this structural vibration using deterministic force models (Mirshekari et al., 2018). In this study, we adopt a probabilistic perspective, assuming individuals do not produce exact identical force-time profiles. However, the walking pattern is similar over different space-time, and the wave profile will exhibit intrinsic and extrinsic randomness as a function of weight, height, structural properties, and background noise, among other factors (Pan et al., 2017). Naturally, a substantial subject database is crucial for such robust statistical characterization. The impact force ($kgms^2$ or Newton (N)) due to footstep impact on the ground is transmitted through the structure and recorded as a vibration signal by a geophone sensor(in Volt(V)). Thus, the recorded signal has induced properties of both the structure and the individual walking pattern.

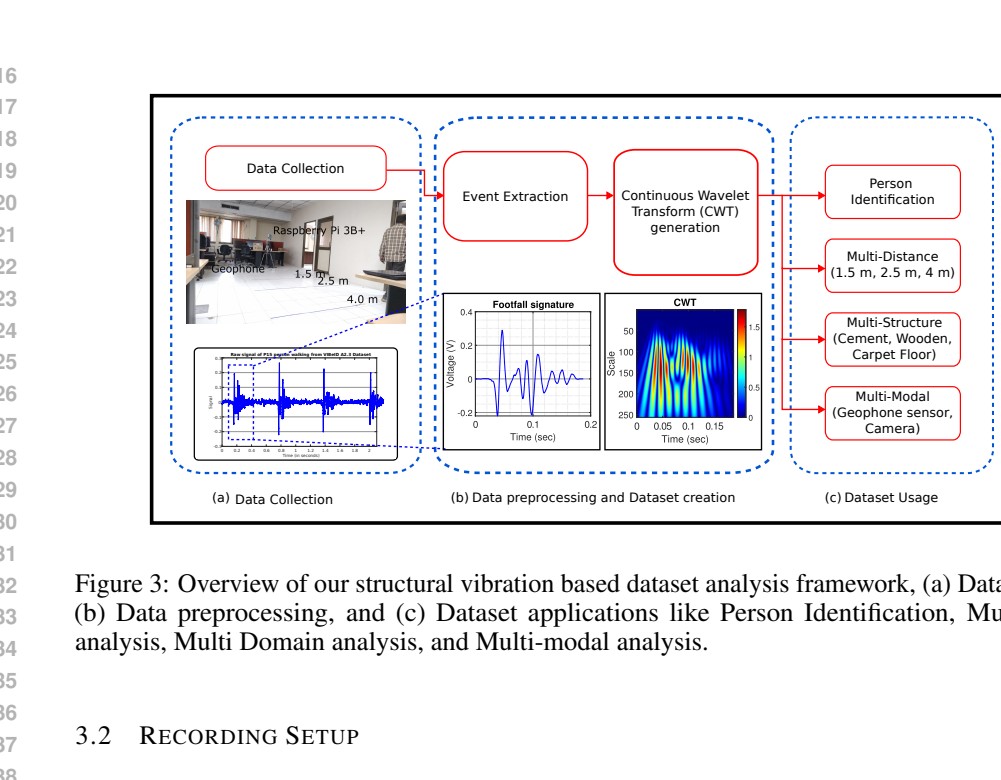

Figure 3: Overview of our structural vibration based dataset analysis framework, (a) Data acquisition, (b) Data preprocessing, and (c) Dataset applications like Person Identification, Multi-Distance analysis, Multi Domain analysis, and Multi-modal analysis.

## 3.2 RECORDING SETUP

A single geophone sensor with a sensitivity of 2.88 V/m/sec and a gain of 10 was used for both indoor and outdoor environments. Vibration signals were captured using a Logic sound card hat equipped with a 16-bit analog-to-digital converter (ADC) operating at a sampling rate of 8 kHz. The sound card interfaced with a Raspberry Pi 3B+ having 1 GB of RAM and 16 GB of storage for indoor data (VIBeID A1, A2, and A3). For outdoor data (VIBeID A4), a single seismic sensor was interfaced with a Sony CXD5602 Spresense embedded microcontroller. Vibration signals were initially recorded at 16 kHz due to hardware limitations and subsequently downsampled to 8 kHz for consistency with the indoor datasets. Details of each recording environment are provided in detail in the supplementary section.

## 3.3 DATASET DETAILS

As shown in figure 2, the sensor, with an effective sensing area of 50-100m² was placed on the floor, and participants were instructed to walk within this region in to-and-fro motion. The data collection protocol excluded any concurrent human activity. Background noise remained unmitigated throughout the recording process. As shown in figure 3, the dataset can be furthur used for four different use-cases. The detail are as follows:

- **VIBeID A1:** Vibration signals were recorded from 100 individuals at a distance of 2.5 m-4.0 m from the sensor. Each individual has 20 minutes of recorded data, totaling 33.66 hours.

- **VIBeID A2:** Data from 30 individuals were collected on a cement floor at three distances from the sensor: 1.5 m (A2.1), 2.5 m (A2.2), and 4.0 m (A2.3). Each individual has 15 minutes of recorded data for each distances, totaling 22.5 hours. To evaluate model performance on different distance range from the sensor, six cross-domain tasks were created: A2.1→A2.2, A2.1→A2.3, A2.2→A2.1, A2.2→A2.3, A2.3→A2.1, and A2.3→A2.2.

- **VIBeID A3:** Data from 40 individuals were collected on wooden (A3.1), carpet (A3.2), and cement (A3.3) floors, at a distance of 2.5 m-4.0 m from the sensor. Each individual has 20 minutes of data per floor, totaling 30 hours. To study the effect of different structure on model performance, we created six cross-domain tasks: A3.1→A3.2, A3.2→A3.3, A3.2→A3.1, A3.2→A3.3, A3.3→A3.1, and A3.3→A3.2.

- **VIBeID A4:** Data from 15 individuals were recorded using single geophone (A4.1) and two off-the-shelve cameras (left camera-A4.2a and right camera-A4.2b). Each individual has 10 minutes of recorded data, totaling 2.5 hours.

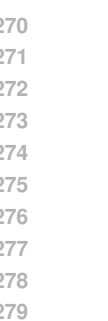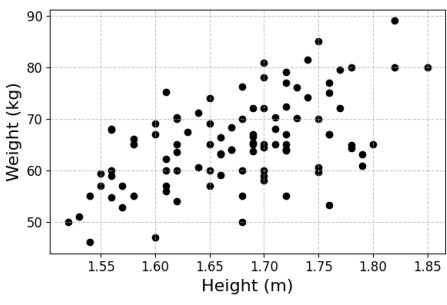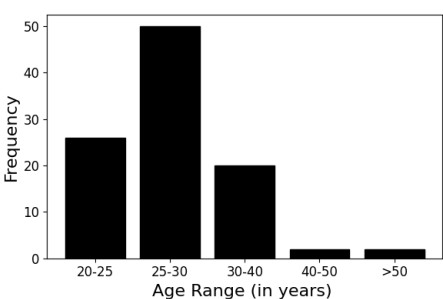

Figure 4: Distribution of Subjects in VIBeID A1, (a) Height and weight distribution of the 100 participants, and (b) Age distribution of the participants in VIBeID A1.

## 3.4 STATISTICAL ATTRIBUTES OF THE DATASET

Table 2: VIBeID Datasets: Sample Details and Characteristics

| Name | | Experiments | Number of Persons | Length of data | Samples |
|---|---|---|---|---|---|
| **VIBeID A1** | | Person Identification | 100 | 33.66 hrs | 1,87,500 |
| **VIBeID A2** | **A2.1** | | | | 33,151 |
| | **A2.2** | Multi-Distance | 2.5 m | 30 | 22.5 hrs | 19,494 |
| | **A2.3** | | 4.0 m | | | 23,030 |
| **VIBeID A3** | **A3.1** | | | | | 63,394 |
| | **A3.2** | Multi-Floor | (1.5 m - 4.0 m) | 40 | 30 hrs | 54,840 |
| | **A3.3** | | | | | 34,328 |
| **VIBeID A4** | **A4.1** | | Geophone | | | 2,171 |
| | **A4.2a** | Multi-Modal | Camera | 15 | 2.5 hrs | 2,670 |
| | **A4.2b** | | | | | 2,074 |

The different statistical attributes of the dataset are given below:

- **Subjects:** In this study, 100 participants, comprising 68 males and 32 females, aged between 20 to 60 years, took part. Details of the participant's gender, age, height, and weight are available in the supplementary section. Each participant was asked to wear flat-bottom shoes that would be comfortable for walking. Table 2, shows the description of the sub-datasets within VIBeID, including sampling details, number of participants, recording duration, their use cases, and environment The data collection process strictly adhered to the rigorous guidelines established by the Institutional Review Board (IRB).

- **Anthropometrics:** As shown in the figure 4, the participants' heights vary from approximately 1.40 m to 1.90 m, and weights range from 40 kg to 90 kg. The dataset's details regarding age, gender, height, and weight are available on the GitHub repository.

- **Data Collection Environment:** Indoor data collection involves multiple 5-minute walking sessions for each participant between 11:00 AM and 6:00 PM.

- **Environmental Noise:** The indoor datasets potentially include noise from typical building operations such as air conditioning. These temporal variations and noises reflect real-world conditions in a multi-functional building environment.

## 3.5 DATA PREPROCESSING

As highlighted by (Pan et al., 2017), footstep vibration can be affected by unknown wave propagation across different locations due to floor heterogeneity. Thus, identifying each footstep event from background noises is an essential goal for data preprocessing. We preprocessed the data using the toolkit provided by (Anchal et al., 2020), which uses unsupervised clustering to extract footstep events based on both statistical features (skewness and kurtosis) and spectral features based on energy bins(40-80Hz, 80-120Hz, and 120-160Hz). We take a rolling window of 375ms with a 50% overlap and calculate 134 features. After preprocessing, we structured the dataset based on footstep events, grouping them so that each row represents one sample consisting of 2-10 footstep events (Anchal et al.,

2020). As shown in the figure 5, we converted the extracted footstep events to 2-D time-frequency images using Continuous Wavelet Transform (CWT) to focus on the changes in energy distribution, which is unique for each and every individual (Xu et al., 2024). Each CWT image is considered as a single sample for deep learning analysis. For the data preprocessing of Dataset VIBeID A4.2,

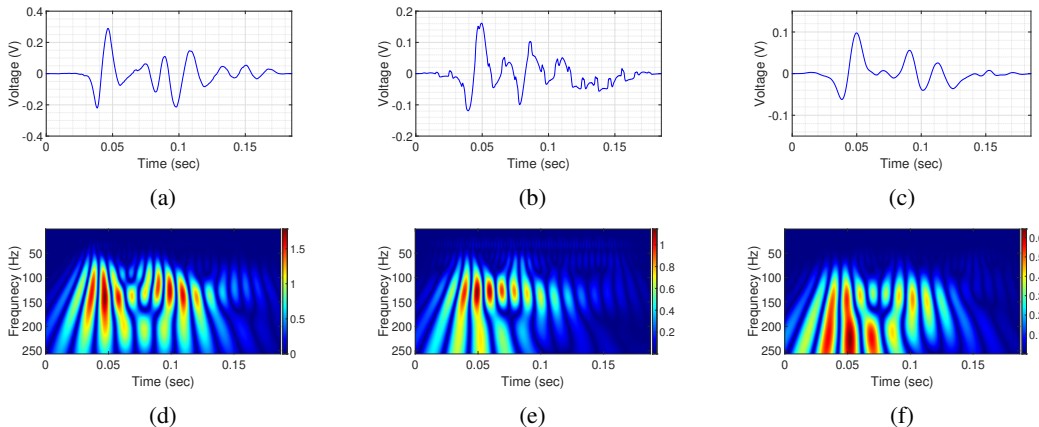

Figure 5: This image depicts vibration signal from three individuals collected from wooden floor: (a) Person 1, (b) Person 50, and (c) Person 100. The corresponding CWT images for each person's footprint are shown in (d), (e), and (f) respectively.

we used the toolkit provided by (Song et al., 2022). We extract Gait Energy Images (GEI) from video footage and classify them using a deep neural network. The process begins with converting the original MP4 video files into individual JPEG images (frames), each maintaining a high resolution of 1920 x 1080 pixels to capture detailed gait information. Next, we use the Mask R-CNN model (pre-trained on the ImageNet dataset) to detect humans within each image frame, allowing us to isolate the human silhouettes from the background (He et al., 2017). This step ensures that the image is normalized based on the extracted silhouette, focusing solely on the relevant body region for gait analysis. Finally, we formulate the GEI by merging the gait cycle images, resulting in a standardized size of 88 x 128 pixels. This consistent size ensures compatibility during subsequent processing and analysis.

## 4 BASELINES EVALUATION OF THE DATASET

### 4.1 PERSON IDENTIFICATION USING STRUCTURAL VIBRATION

This work investigates person identification using structural vibration data as a multi-class classification problem. Each dataset is treated as a separate standalone dataset for this analysis.

**Baselines:** The details of the two main approaches are as follows:

- Machine Learning Approach: We use the toolkits provided by (Anchal et al., 2020), which utilize 134 pre-defined features, to represent the gait patterns within the data. These features are extracted from the raw vibration signals. We then create labeled datasets from the features and perform five-fold cross-validation for robust evaluation. We investigate the performance of Support Vector Machines with radial basis function (SVM-RBF) and Random Forest(RF). The hyperparameters for SVM-RBF C and $\gamma$ were set to $10^7$ and $10^{-7}$. In RF, we used entropy as the splitting criteria, and depth is set at 30.

- **Deep Learning Approach:** We utilize ResNet-18 and ResNet-50 as backbone architectures (He et al., 2016) for two evaluation tasks:

  - *Multi-class classification:* Training ($D_{\text{train}}$) and testing ($D_{\text{test}}$) sets are disjoint at the sample level ($D_{\text{train}} \cap D_{\text{test}} = \emptyset$). Models are trained using either single-image input (S) or five consecutive images (M). For multi-image input, a Conv2D layer (15 channels,

Table 3: Accuracy (%) of datasets evaluated using machine learning methods. Accuracy is in "mean (std.)" format.

| Classifier | Dataset / Footstep Events | 2 | 5 | 7 | 10 |
|---|---|---|---|---|---|
| Random Forest | A1 | 81.50(4.49) | 88.10(4.56) | 90.54(4.33) | 92.67(3.91) |
| | A2.1 | 51.19(1.54) | 61.89(2.77) | 65.60(2.69) | 69.37(2.19) |
| | A2.2 | 59.69(3.65) | 68.15(4.00) | 69.32(4.67) | 72.35(6.49) |
| | A2.3 | 59.93(2.33) | 71.87(2.30) | 74.87(3.14) | 76.31(2.90) |
| | A3.1 | 81.67(3.79) | 88.71(3.28) | 91.10(3.33) | 93.60(2.69) |
| | A3.2 | 85.21(2.92) | 92.29(2.46) | 94.369(2.78) | 95.87(1.97) |
| | A3.3 | 81.01(3.81) | 88.99(3.55) | 91.58(3.46) | 93.67(2.89) |
| | A4.1 | 76.57(3.89) | 79.47(2.72) | 82.87(2.77) | 81.04(5.43) |
| SVM (RBF) | A1 | 32.35(1.24) | 44.02(2.21) | 46.42(1.98) | 51.97(1.76) |
| | A2.1 | 79.88(4.80) | 87.99(4.14) | 91.08(3.59) | 92.51(3.41) |
| | A2.2 | 84.33(4.57) | 89.49(4.26) | 92.64(3.73) | 94.31(3.45) |
| | A2.3 | 86.66(2.32) | 94.25(1.14) | 95.66(1.65) | 97.03(1.48) |
| | A3.1 | 37.83(1.88) | 45.71(2.60) | 50.47(2.90) | 56.71(3.51) |
| | A3.2 | 63.87(2.24) | 73.79(2.45) | 77.048(2.43) | 80.01(2.72) |
| | A3.3 | 61.58(2.45) | 71.31(3.64) | 74.35(4.42) | 76.20(3.87) |
| | A4.1 | 33.14(2.08) | 37.28(3.09) | 37.13(7.83) | 44.99(4.89) |

kernel size = 7, stride = 2, padding = 3) is added before the backbone, feeding into the base network (output channels = 64 for ResNet-18/50). The final dense layer matches the number of dataset classes, with categorical cross-entropy loss and Adam optimizer. Data is split into 80% training and 20% testing.

- *Person identification:* Training ($C_{train}$) and testing ($C_{test}$) classes are disjoint ($C_{train} \cap C_{test} = \emptyset$), evaluating model generalization using top-$k$ accuracy and precision. We employ a triplet loss (margin = 1.0) to reduce intra-class distances and increase inter-class separability, with dynamically generated triplets during training. The backbone is modified to produce 512-dimensional embeddings by replacing the classification head with a linear layer.

For both tasks, models are trained for up to 25 epochs with a batch size of 16 experiments are repeated five times, reporting the mean accuracy and standard deviation.

Table 4: Classification accuracy (%) of datasets evaluated using deep learning methods. Accuracy is in "mean (std.)" format.

| Datasets | ResNet50 (S) | ResNet50 (M) | ResNet18 (S) | ResNet18 (M) |
|---|---|---|---|---|
| A1 | 84.45 (1.10) | 88.96 (0.32) | 81.29 (3.41) | **92.46** (0.43) |
| A2.1 | 84.68 (2.35) | **86.09** (3.08) | 82.17 (1.02) | 84.60 (2.83) |
| A2.2 | **87.70** (3.70) | 85.88 (1.92) | 68.50 (4.03) | 74.58 (1.54) |
| A2.3 | **87.73** (0.93) | 86.81 (2.20) | 70.14 (2.64) | 73.26 (3.15) |
| A3.1 | 87.25 (2.75) | 91.99 (0.78) | 87.35 (0.84) | **92.35** (0.21) |
| A3.2 | 82.61 (0.50) | 89.48 (1.34) | 84.65 (2.46) | **89.89** (1.89) |
| A3.3 | 89.38 (3.20) | 88.34 (3.94) | **90.52** (1.20) | 88.19 (0.13) |
| A4.1 | 93.70 (1.67) | 84.87 (2.51) | **94.71** (3.61) | 90.79 (2.91) |

Table 5: Identification Accuracy (%) of datasets evaluated using multi-input models (Pretrained-P, None-N). Accuracy is in "mean (std.)" format.

| Model | A1 | A2.1 | A2.2 | A2.3 | A3.1 | A3.2 | A3.3 | A4.1 |
|---|---|---|---|---|---|---|---|---|
| ResNet-18 (P) | **78.10** (1.42) | **81.53** (2.19) | **91.75** (1.87) | **92.67** (0.57) | **75.13** (2.34) | **84.09** (0.92) | 84.43 (2.78) | **84.31** (1.12) |
| ResNet-18 (N) | 66.62 (4.87) | 74.61 (3.21) | 78.27 (2.18) | 76.91 (3.45) | 68.56 (4.02) | 70.04 (3.89) | **85.38** (4.33) | 82.35 (2.01) |
| ResNet-50 (P) | 69.87 (2.65) | 72.90 (1.13) | 75.70 (2.84) | 84.50 (0.72) | 62.17 (2.21) | 65.80 (1.91) | 73.82 (0.47) | 80.31 (2.53) |
| ResNet-50 (N) | 58.70 (3.77) | 68.50 (1.92) | 73.61 (2.61) | 64.54 (3.14) | 67.66 (2.89) | 63.72 (4.25) | 71.46 (2.03) | 70.59 (1.98) |

**Results:** A comparative analysis of machine learning methodologies on the datasets (Table 3) revealed superior performance in indoor environments. This suggests that the hand-crafted features used by

the toolkits are better optimized for indoor footstep characteristics. Random Forest achieved better performance than SVM (RBF kernel) except multi-distance datasets. Using multiple footstep inputs improved recognition results by $\sim 5 - 10\%$, this is most likely due to the ability of multiple footsteps to capture a more comprehensive representation of the underlying walking patterns. An analysis of deep learning methodologies is presented in Table 4, for each datasets. Since pre-trained models for structural vibration signals are unavailable, all models were trained from scratch. Both ResNet-18 and ResNet-50 demonstrated equally good performance. This is likely attributable to applying a fixed hyperparameter evaluation rather than individual fine-tuning for each dataset. The higher accuracy of the outdoor dataset shows that deep learning-based methods are more suitable for applications in varied and dynamic environments.

Table 6: Accuracy (%) for domain adaptation. Accuracy is in "mean (std.)" format.

(a) VIBeID A2 Multi-Distance Datasets

| Methods | A2.1→A2.2 | A2.1→A2.3 | A2.2→A2.1 | A2.2→A2.3 | A2.3→A2.1 | A2.3→A2.2 |
|---|---|---|---|---|---|---|
| Source-Only | 21.20 (1.69) | 17.15 (1.50) | 19.14 (3.86) | 9.35 (3.62) | 10.57 (2.12) | 13.35 (2.43) |
| Fine-Tuning(3-layers) | 82.98 (2.03) | 81.34 (1.95) | 68.15 (2.36) | 75.95 (2.45) | 66.62 (2.82) | **73.04** (2.41) |
| Fine-Tuning | **85.67** (2.11) | **84.26** (2.43) | **72.28** (2.76) | **76.61** (2.27) | **71.36** (3.71) | 72.51 (2.59) |

(b) VIBeID A3 Multi Floor Datasets

| Methods | A3.1→A3.2 | A3.1→A3.3 | A3.2→A3.1 | A3.2→A3.3 | A3.3→A3.1 | A3.3→A3.2 |
|---|---|---|---|---|---|---|
| Source-Only | 2.00 (1.12) | 3.12 (2.53) | 3.8 (0.92) | 5.05 (1.70) | 3.41 (1.24) | 7.40 (2.50) |
| Fine-Tuning(3-layers) | 75.36 (1.20) | **76.70** (0.46) | 82.14 (2.4) | 79.64 (3.40) | 80.24 (1.02) | 80.13 (1.05) |
| Fine-Tuning | **80.00** (0.50) | 76.60 (2.03) | **86.03** (1.00) | **79.79** (3.20) | **85.24** (1.21) | **82.70** (1.80) |

## 4.2 Domain Adaptation on different distance and floor types

Building on Section 2.1, we address the limitations of existing methods that assume consistent data distributions between training and testing. We explore person identification using deep learning on 12 cross-domain datasets (6 from VIBeID A2, and 6 from VIBeID A3). We present results for both the source domain and the fine-tuned models.

**Baselines:** We use only the ResNet18 (M) for domain adaptation use-case because of its fast computation compared to other baseline models. We use a 70-20-10 split for the train-test-validation process, where $70\%$ of data are used for source model training, $20\%$ of target data is used for fine-tuning and $10\%$ of target data is used for model evaluation. Our approach are as follows:

- Source-Only: This baseline approach trains a model solely on data from the source domain ($70\%$ Source data) and is tested on the target domain ($10\%$ target domain). This represents the lower bound of performance achievable in our domain adaptation task.

- Fine-tuning with limited Target Data: Following standard transfer learning principles, we fine-tuned a pre-trained source model (trained on $70\%$ Source data) with limited labeled data from the target domain ($20\%$ target domain) and evaluated the performance ($10\%$ target domain). We experimented with two fine-tuning approaches: a) unfreezing and fine-tuning the last three layers, and b) unfreezing the entire model. The hyper-parameters remained consistent with those used for training the source model.

**Results:** Analysis of Table 6 shows that both multi-distance (Table 6a) and multi-floor (Table 6b) data perform well during fine-tuning with limited labeled data, indicating the approach's effectiveness in adapting to domain variations. For scenarios with limited computational resources, fine-tuning with only the last three layers unfrozen is a viable alternative, offering faster training time while maintaining comparable accuracy. We observe variations in model performance across structures (multi-floor) and sensor distances (multi-distance). This likely arises from (a) material non-homogeneity, and (b) background noise. Despite these challenges, fine-tuning with limited data has achieved accurate person identification using standard classification methods.

Table 7: Identification accuracy (%) and Mean Average Precision (mAP) comparison for multi-modal datasets ($A1^s$, A4.1, A4.2a, and A4.2b).

| Models | ResNet-18 | | | | ResNet-50 | | | |
| | None | | Pre-trained | | None | | Pre-trained | |
| | Accuracy | mAP | Accuracy | mAP | Accuracy | mAP | Accuracy | mAP |
| --- | --- | --- | --- | --- | --- | --- | --- | --- |
| A4.1 | 84.31 | 0.7655 | 82.35 | **0.8450** | 66.67 | 0.6444 | **86.27** | 0.7105 |
| $A1^s$ | 83.74 | 0.6628 | 80.33 | **0.7364** | 78.08 | 0.6298 | **87.44** | 0.6392 |
| A4.2a | **82.86** | **0.3713** | 77.48 | 0.3342 | 38.66 | 0.2485 | 65.21 | 0.3344 |
| A4.2b | **82.05** | **0.4243** | 78.80 | 0.4214 | 47.13 | 0.2565 | 75.38 | 0.3801 |

### 4.3 MULTI-MODAL ANALYSIS

**Baselines:** Our goal is to compare two modalities aka structural vibration-based spectograms and Gait Energy Image (GEIs), for person identification. We compute the GEI for the camera modality and treat each GEI as a sample, regardless of viewpoint. We also use a subset of the VIBeID A1 dataset, denoted as VIBeID $A1^s$, containing only individuals whose data was also recorded in the outdoor environment. Spectrograms used are ten consecutive footstep events.

**Results:** Table 7 presents identification accuracy (%) comparisons across four datasets: $A1$, $A1^s$ (spectrograms), and $A4.2a$, $A4.2b$ (Gait Energy Images captured from two camera perspectives). For spectrogram datasets, ResNet-50 with pre-trained weights achieves the highest accuracy on $A1^s$ (87.44%), reflecting a 3.7% improvement over the best ResNet-18 configuration (83.74%) and a 20.77% improvement over ResNet-50 without pre-training (66.67%). $A1^s$ consistently outperforms $A1$, with up to a 1.17% improvement (87.44% vs. 86.27%) under the best pre-trained ResNet-50 configuration, indicating the positive impact of enhanced preprocessing.

When considering the mean Average Precision (mAP), spectrogram datasets exhibit superior results, especially under pre-trained models. For $A4.1$, pre-trained ResNet-18 achieves the highest mAP (0.8450), while $A1^s$ with pre-trained ResNet-50 closely follows (0.7364), further highlighting the advantage of pre-training. GEI datasets, on the other hand, report significantly lower mAP values. The highest mAP for $A4.2a$ is only 0.3713 (ResNet-18, no pre-training), and $A4.2b$ achieves 0.4243 (ResNet-18, no pre-training). This discrepancy suggests that spectrograms are inherently better suited for accurate and reliable identification in this experimental framework, while GEIs suffer from increased sensitivity to cross-view variations and limited discriminative capacity, particularly in mAP performance.

## 5 LIMITATIONS AND CONCLUSION

This work establishes a foundation for exploring structural vibrations in soft biometrics for person identification. We are committed to actively maintaining and expanding both the VIBeID dataset and its benchmark. we recognize opportunities for growth, such as increasing the number of participants and exploring a broader range of environments, we view these as exciting challenges. Our goal is to enhance the dataset's robustness by including more individuals and expanding into diverse domains, incorporating varied environments, materials, and conditions to unlock new possibilities and drive innovation in this field.

VIBeID offers a significant leap forward in data volume, containing over 88.66 hours of vibration recordings and also provides a baseline using established methods. We anticipate open-sourcing the dataset will accelerate future research to explore more efficient approaches. VIBeID can serve as a springboard for significant advancements in the field of deep learning and structural vibration based person identification. Beyond person identification, VIBeID's value extends to structural vibration analysis. The metadata, including subject age, gender, height, weight, and structural details, is invaluable for researchers exploring applications in bio-mechanics, where understanding the relationship between body characteristics and structural vibration is crucial.

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

## 6 APPENDIX

The supplementary materials consist of:

- Details of the rooms used in the dataset.
- Statistical attributes of the dataset.
- Procedures for data collection and preprocessing.
- Details of the baseline implementations.
- Limitations, discussions, and potential impact.
- Analysis on variation of speed.
- URL to data and miscellaneous information.

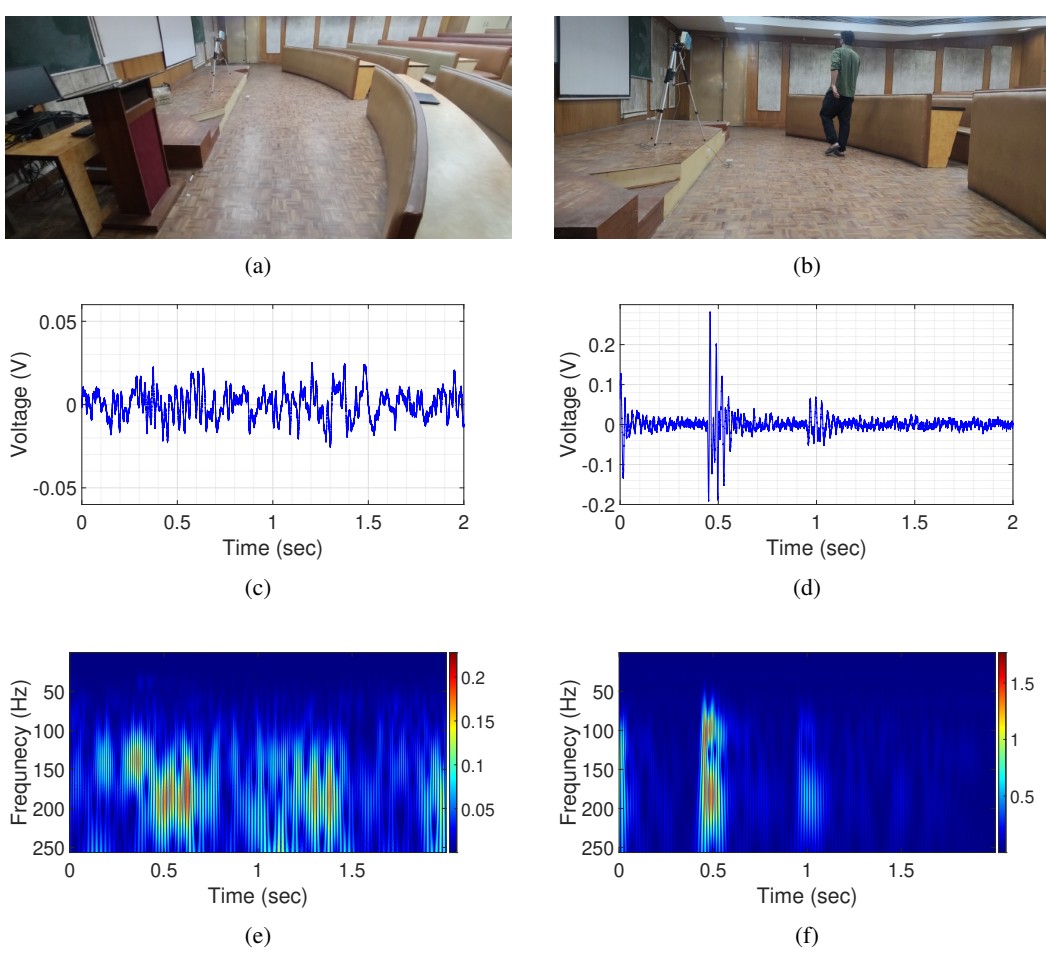

Figure 6: (a) Empty wooden floor room, (b) Participant walking in front of the sensor during data collection, (c) Raw noise signal acquired from the geophone sensor, (d) Raw vibration signal recorded from participant P14 walking, (e) Continuous Wavelet Transform (CWT) of the noise signal, (f) CWT of the P14 walking signal, highlighting the transformed features.

### 6.1 LOCATION DETAILS

#### 6.1.1 WOODEN FLOOR

The data collection environment is a ground-floor classroom with wooden flooring. The walking area measured 50 -100 $m^2$. Raspberry Pi 3B+ was placed in the podium, while a geophone sensor was

placed on the floor. This room is used for data collection of dataset VIBeID A1, and A3.1. Figure 6 shows a photo of the room, a sample of noise distribution within the room, and a visualization of data collected from a walking person.

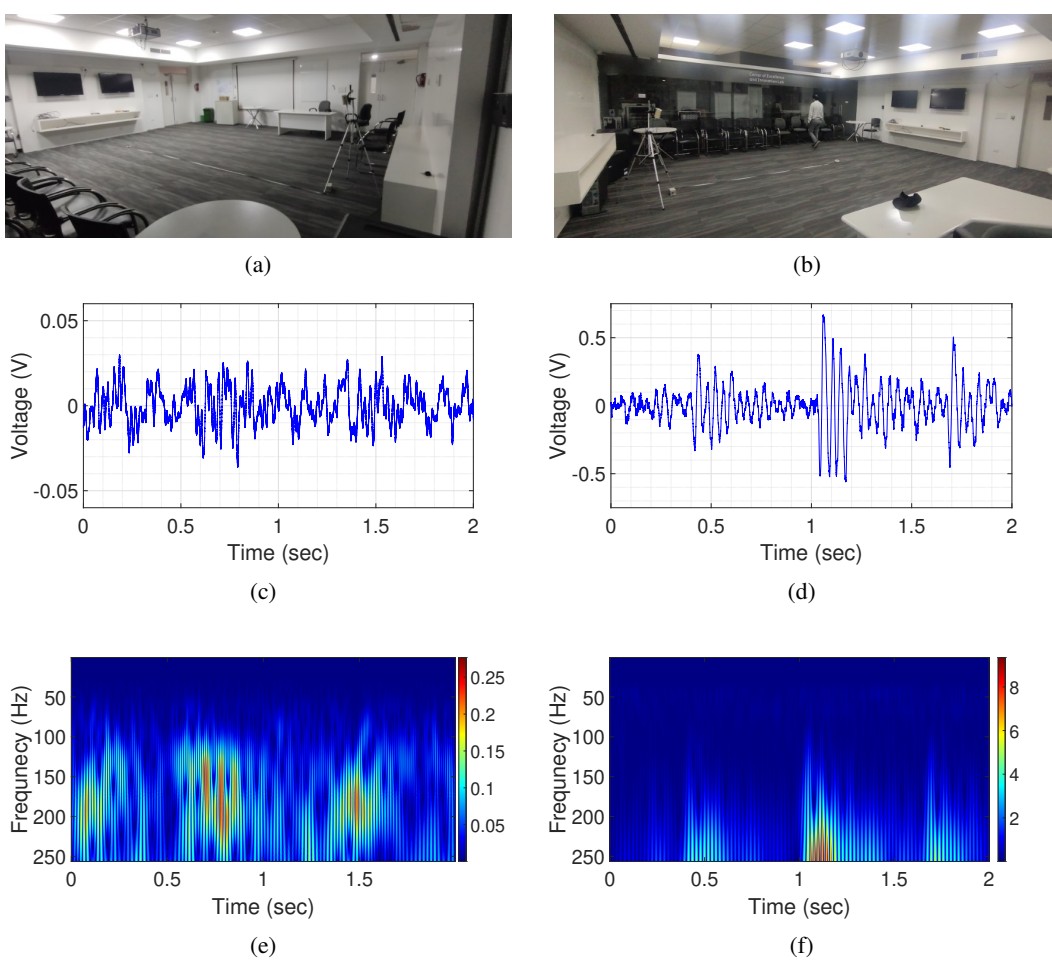

Figure 7: (a) Empty carpet floor room, (b) Participant walking in front of the sensor during data collection, (c) Raw noise signal acquired from the geophone sensor, (d) Raw vibration signal recorded from participant P15 walking, (e) Continuous Wavelet Transform (CWT) of the noise signal, (f) CWT of the P15 walking signal, highlighting the transformed features

### 6.1.2 CARPET FLOOR

The room is a ground-floor conference room with carpeted flooring overlying concrete. The thickness of the carpet is 9mm. The room is furnished with chairs and a table. Participants are asked to walk within a radius of 50-100 $m^2$. This room is used for data collection of dataset VIBeID A3.2, where 40 person walked for 15 minutes each. The figure 7 illustrates the data collection setup for VIBeID A3.2, including a room photo, noise distribution sample, and walking person data visualization.

### 6.1.3 CEMENT FLOOR

The room is a research lab on the third floor, covered in tiles. The room measures 50-100 $m^2$ as walking area. It is an active research lab with regular activities occurring in the background. This room is used for data collection of dataset VIBeID A3.3, and VIBeID A2 (1.5 m, 2.5 m, and 4.0 m), to study the effect of sensing distance and how effective our system can be when people walk far

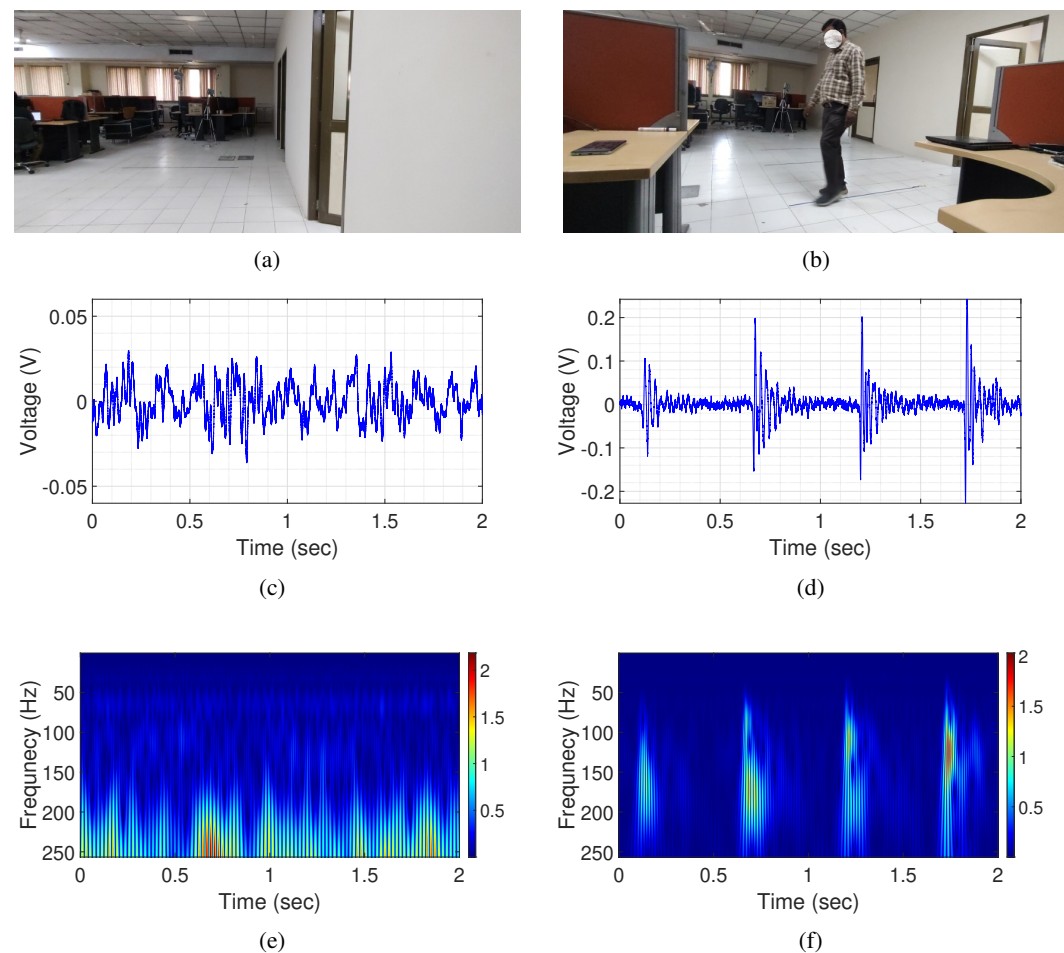

Figure 8: (a) Empty cement floor room, (b) Participant walking in front of the sensor during data collection, (c) Raw noise signal acquired from the geophone sensor, (d) Raw vibration signal recorded from participant P15 walking, (e) Continuous Wavelet Transform (CWT) of the noise signal, (f) CWT of the P14 walking signal, highlighting the transformed features.

away from the sensors. Figure 8 shows a photo of the room, a sample of noise distribution within the room, and a visualization of data collected from a walking person.

### 6.1.4 OUTDOOR GROUND

The data collection environment is an open outdoor playground where the vibration signals of 15 individuals were recorded. Due to the open nature of the space, extraneous noise from activities such as people walking, running, and playing cricket was unavoidable. While we tried to minimize background movement within the 50-100 $m^2$ sensing area, the camera's wide field of view did capture some additional human activity in the video data. While this presented an initial challenge for extracting clean human figures (due to residual background noise), it also offered a valuable opportunity to test the robustness of the GEI formation process under slightly non-uniform conditions. We opted for outdoor data collection for two primary reasons. First, to establish a control group as all previous data recordings were conducted indoors. Secondly, the outdoor data collection served as a foundation for comparative analysis of our method against established soft biometric techniques, such as gait recognition, in a more realistic setting. Open ground is used for data collection of two modalities, structural and vision based; the structural based vibration collected data named as VIBeID A4.1 and vision based collected data is used to formulate VIBeID A4.2. Figure 9 shows the

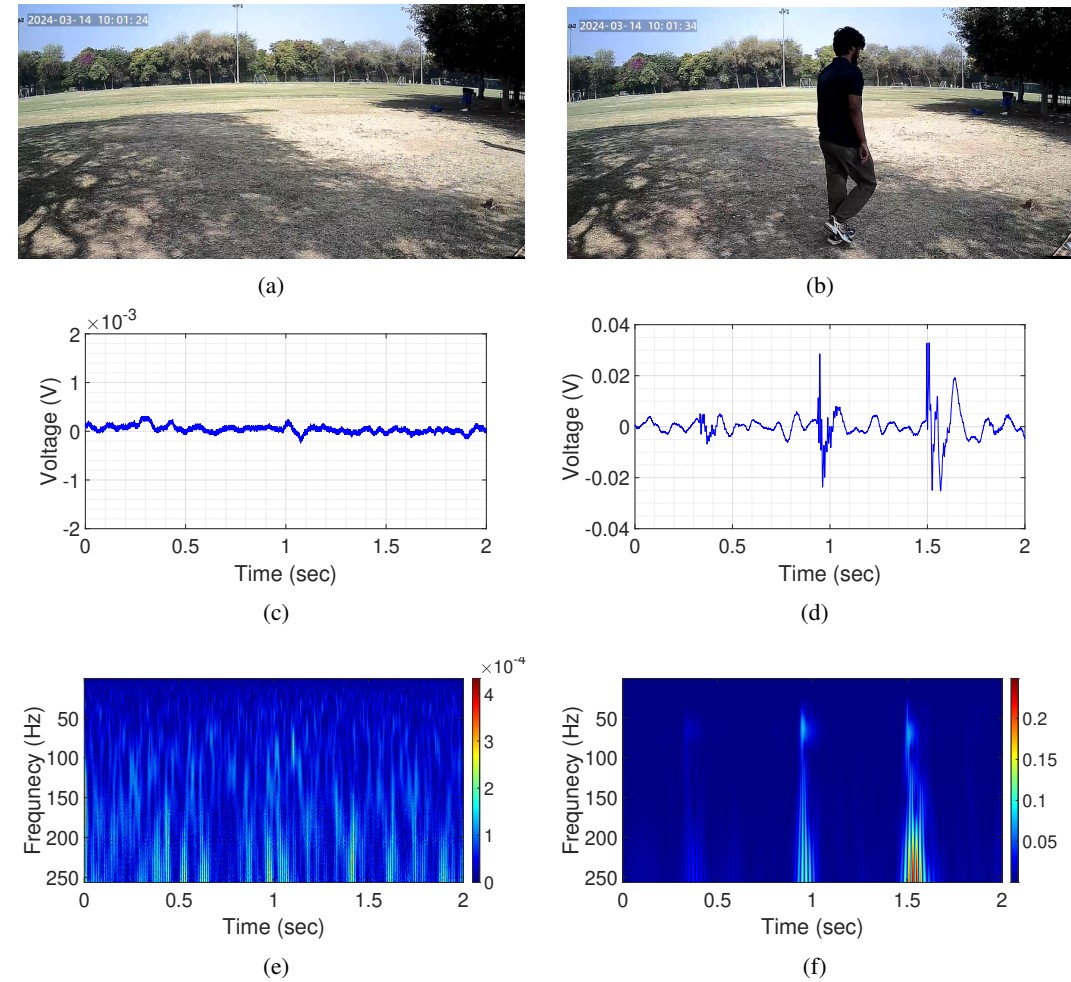

Figure 9: (a) Empty outdoor ground, (b) Participant walking in front of the sensor during data collection, (c) Raw noise signal acquired from the geophone sensor, (d) Raw vibration signal recorded from participant P9 walking, (e) Continuous Wavelet Transform (CWT) of the noise signal, (f) CWT of the P9 walking signal, highlighting the transformed features.

location setup, a sample of noise distribution and a visualization of data collected from a walking person.

### 6.1.5  OTHER COMMENTS

Figure 4-7(c), clearly demonstrates the variation in noise distribution across different rooms and outdoor environment. Notably, the signal-to-noise ratio (SNR) exhibits significant dependence on the floor level, with ground floor locations typically having high SNR compared to the third floor. Additionally, we observed that the amplitude of the source signal exhibits an inverse relationship with distance from the sensor. To achieve effective event extraction from the signal, scenario-specific training data is used.

## 6.2  EXPERIMENTAL SETUP AND DATA COLLECTION PROTOCOL

### 6.2.1  HARDWARE DETAILS

We collected structural vibration data, with a single geophone for Multiple Indoor (VIBeID A1, A2, A3) and outdoor (VIBeID A4) environment, along with two cameras.

- **Geophone :** Geophone is a sensor that converts the ground movement into voltage, which can be recorded by using any microcontroller or microprocessor having an analog-to-digital converter (ADC). The used geophone has a sensitivity of 2.88 V/m/sec and a pre-amplification gain of 10.

- **Raspberry Pi 3B+ :** For indoor experiments (VIBeID A1, A2, and A3), we have used raspberry Pi 3B+ featuring BCM2837B0 64-bit ARM-based Cortex-A53 processor running at 1.4 GHz with 1 GB of RAM and 16 GB of storage. The geophone sensor was interfaced with raspberry Pi 3B+ using a logic sound card hat equipped with a 16-bit analog-to digital converter (ADC) at a sampling rate of 8 KHz.

- **Sony spresense Board :** In our outdoor experiments (VIBeID A4.1), we used the Sony CXD5602 Spresense board. This board features an ARM Cortex-M4F processor with six cores running at 156 MHz and a 16-bit A/D conversion output. We opted for the Sony Spresense board due to its low power consumption—only 1 W (5 V @ 200 mA)—compared to the Raspberry Pi 3B+, which requires approximately 7.5 W (5 V @ 1.5 A). This energy efficiency makes the Sony Spresense board well-suited for outdoor data collection experiments. However, it has a limitation: it can only record signals at 16 KHz and 32 KHz. Therefore, we initially recorded vibrational signals at 16 KHz and subsequently downsampled the data to 8 KHz to maintain consistency with the indoor datasets (VIBeID A1, A2, and A3).

- **Camera :** For (VIBeID A4.2) dataset, we have used 2 cameras that have a CMOS sensor (0.84667 cm) with 3 MP, $95°$ viewing angle, with frame rate of 20 fps. It supports Wi-fi protocol to remotely view the recorded video and save it to smartphone or cloud services.

Figure 10 shows the images of geophone sensor, raspberry pi 3B+, and camera.

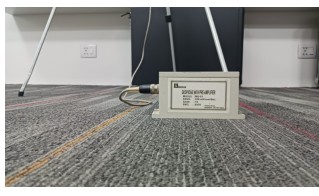
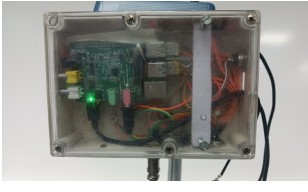
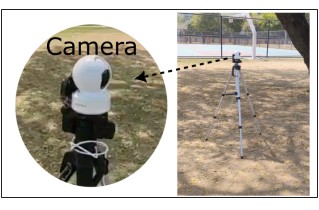

| (a) Geophone | (b) Raspberry Pi 3B+ | (c) Camera |

Figure 10: Hardware Components: (a) Geophone sensor for capturing structural vibrations, (b) Raspberry Pi 3B+ board for data acquisition, and (c) Camera for outdoor data collection.

### 6.2.2 DATA COLLECTION PROTOCOL

Prior to data collection, participants were briefed for 5 minutes, about the task to ensure clear understanding. We instructed the participants to complete a repetitive walking task. They were asked to walk from a designated starting point (Point A) to another designated endpoint (Point B) at a natural pace, turn around at Point B, and then repeat the walk back to Point A. Each session lasted for five minutes. Multiple such sessions were conducted, depending on the sub-task. Participants were informed that they could stop and rest at any point during the session (if they feel uncomfortable); the recording would be paused and restarted upon resuming the walk. Apart from intended walking activity any additional human activities were strictly prohibited. This controlled environment minimized background noise and ensured the capture of pure walking patterns. It is important to note that while we focused on minimizing human activity-related noise, structural vibrations from non-human activities were not controlled.

### 6.3 DATA PRE-PROCESSING TOOLKIT

The raw vibration signal undergoes data pre-processing (Figure 11), where we extract events from the raw signal. This events are further converted to CWT images. We have used the toolkit provided by (Anchal et al., 2020) to extract footstep events. It uses Gaussian Mixture Model (GMM), as an unsupervised clustering technique to extract footstep events from noise. Each recording is divided into equal parts using a sliding window approach (375 ms), and we extract features from this window (see Table 8). The steps are as follows:

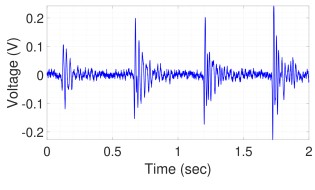 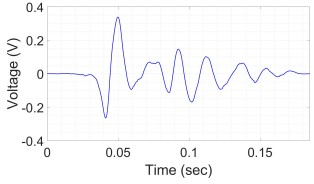 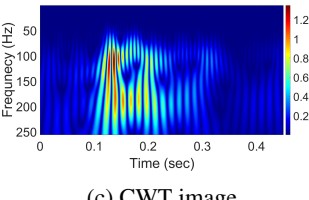

(a) Raw vibration signal  (b) Pre-processed signal  (c) CWT image

Figure 11: Data Processing Steps: (a) Raw vibration signal acquired from the geophone sensor. (b) Pre-processed signal after noise reduction and event segmentation. (c) Extracted events converted into a CWT image for further analysis.

- **Modeling Footstep Events** : Each feature vector, $f_i$, is modeled as:

$$\sum_{c=1}^{C} \phi_m \cdot \mathcal{N}(f_i|\mu_m, \Sigma_m) \tag{1}$$

  where,
  - $C$ (number of clusters) is set to 2
  - $N$ is the number of training samples
  - $\phi_m$ is the weight of the $m^{th}$ cluster (represents its probability)
  - $\mu_m$ is the mean vector of the $m^{th}$ cluster
  - $\Sigma_m$ is the covariance matrix of the $m^{th}$ cluster

- **Training the GMM** : We train the GMM by maximizing the log-likelihood of the training data:

$$\ln p(\mathbf{F}|\Theta) = \sum_{i=1}^{N} \ln \left( \sum_{m=1}^{2} \phi_m \mathcal{N}(f_i|\mu_m, \Sigma_m) \right) \tag{2}$$

  where,
  - $\mathbf{F} = [\mathbf{f}_1^T, \mathbf{f}_2^T, \ldots, \mathbf{f}_N^T]^T$ is the feature matrix
  - $\Theta = \{\phi_1, \phi_2, \mu_1, \mu_2, \Sigma_1, \Sigma_2\}$ [$\Theta$ is maximised by log-likelihood using the EM algorithm]

- **Classifying Footsteps**: The GMM generates two clusters, $C_1$ and $C_2$. We assign labels based on the determinants value (absolute value of the determinant) of the covariance matrices:

$$\text{Cluster} = \begin{cases} \mathcal{E} \to C_1, \hat{\mathcal{E}} \to C_2 & : |\Sigma_1| > |\Sigma_2| \\ \mathcal{E} \to C_2, \hat{\mathcal{E}} \to C_1 & : |\Sigma_2| > |\Sigma_1| \end{cases} \tag{3}$$

  where,

  - where the event and noise class are represented by $\mathcal{E}$ and $\hat{\mathcal{E}}$, respectively,
  - $\Sigma_m$ is the determinant of the covariance matrix of the $m$-th cluster
  - $\hat{\mathcal{E}}$ are parameterized by sets $(\phi_1, \mu_1, \Sigma_1)$ and $(\phi_2, \mu_2, \Sigma_2)$ when $|\Sigma_1| > |\Sigma_2|$

Table 8: Features used for event detection

| Statistical Features | | Spectral Features (Energy bins) | | |
|---|---|---|---|---|
| $f_i^1$ | $f_i^2$ | $f_i^3$ | $f_i^4$ | $f_i^5$ |
| Std. | Kurtosis | 40-80 Hz | 80-120 Hz | 120-160 Hz |

After training the GMM model, we extract footsteps events from each recordings. We train the GMM model using recording of a single person (i.e. person 14's data is used for training model VIBeID A1). Note that changing the training data will affect the GMM model and the number of events extracted. The resulting datasets (pre-processed signal dataset) each has row representing a single footstep event, containing 1500 data points. This pre-processed dataset has been shared for replication of results. The pre-processed event data is simply a denoised signal which exhibits a clearer representation of the underlying events compared to the raw signal.

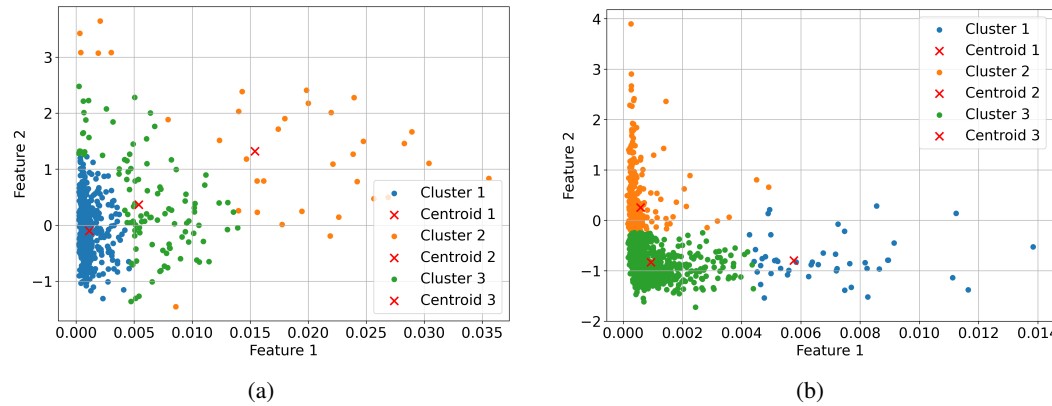

(a)             (b)

Figure 12: Clustering results using a Gaussian Mixture Model (GMM): (a) clustering for concurrent non-human activities, and (b) clustering for concurrent human activities. Data points are partitioned into three clusters—Cluster 1 (blue), Cluster 2 (orange), and Cluster 3 (green). Cluster centroids are represented by red 'X' markers.

## 6.4 WILD SET EVALUATION INVOLVING CONCURRENT HUMAN AND NON-HUMAN ACTIVITY

To evaluate scenarios involving concurrent human and non-human activities, we conducted additional experiments that yielded an additional 30 minutes of data. We named it "Wild set". For such mixed-activity data, we modified the Gaussian Mixture Model (GMM) to three clusters, denoted as $\mathcal{C}_i$ where $i \in \{1, 2, 3\}$. Each cluster $\mathcal{C}_i$ is parameterized by its mixing coefficient $\pi_i$, mean vector $\mu_i$, and covariance matrix $\Sigma_i$. The decision process relies on the determinant of the covariance matrices, $|\Sigma_i|$, which provides a measure of the spread or variance of the cluster in the feature space. The following procedure is adopted:

1. For each cluster $\mathcal{C}_i$ ($i \in \{1, 2, \ldots, k\}$), compute the determinant of its covariance matrix, denoted as $|\Sigma_i|$, where:
$$|\Sigma_i| = \det(\Sigma_i),$$
and $\Sigma_i \in \mathbb{R}^{d \times d}$ is the covariance matrix of cluster $\mathcal{C}_i$ in $d$-dimensional feature space.

2. Identify the two clusters, $\mathcal{C}_{\max1}$ and $\mathcal{C}_{\max2}$, corresponding to the largest and second-largest covariance determinants:
$$|\Sigma_{\max1}| = \max(|\Sigma_i|), \quad |\Sigma_{\max2}| = \max(|\Sigma_i| \setminus |\Sigma_{\max1}|),$$
where $|\Sigma_{\max1}|$ and $|\Sigma_{\max2}|$ are associated with complex noise or human activities due to their higher variability in the feature space.

3. Extract the signal segments corresponding to the clusters, $\mathcal{C}_{\max1}$ and $\mathcal{C}_{\max2}$, where $j \neq \{\max1, \max2\}$. Convert the footstep events into spectrogram representations.

This iterative comparison ensures that clusters with higher variance are robustly separated from those with lower variance, enabling segmentation of human and non-human activities. As shown in Figure 12, the data points are grouped into three distinct clusters within the feature space. The horizontal axis (*Feature 1*) and vertical axis (*Feature 2*) represent extracted features derived from statistical and spectral analysis of the input signal. The centroids of each cluster, indicated as red crosses, demonstrate the separation between noise, human activities, and non-human activities, ensuring dynamic adaptability to mixed data scenarios.

### 6.4.1 EVALUATION SETUP

**Model Configuration :** A pretrained ResNet-18 (IMAGENET1K_V1) (S) was modified by replacing the final fully connected layer with a custom embedding layer of size 512. This enabled the model to extract latent features of spectrogram data.

**Data Preparation :**   We considered data from two individuals. Spectrogram data was organized into two categories:

- **Pure Data:** Spectrograms collected from VIBeIDA1, representing a clean dataset, with normal background noise.

- **Activity Data:** Spectrograms extracted using an unsupervised event detection method based on Gaussian Mixture Models (GMM) with $k = 3$, incorporating either non-human or human noise.

- **Group Activity Data:** Spectrograms obtained through an unsupervised event detection strategy employing Gaussian Mixture Models (GMM) with $k = 3$, capturing ambient noise from two individuals and person-of-interest walking in the vicinity.

**Statistical Testing :**   A two-sample t-test was performed to quantify the difference between the embeddings of pure and noisy clusters. The embeddings were flattened into 1D arrays, and the t-statistic and p-value were computed to evaluate statistical significance.

Table 9: Statistical Test Results for Pure vs. Activity Data

| Comparison | T-Statistic | P-Value |
|---|---|---|
| Pure & Non-Human Activity Data | $-0.772$ | 0.440 |
| Pure & Human Activity Data | $-1.750$ | 0.080 |
| Pure & Group Activity Data | $-1.329$ | 0.183 |
| Pure & Random Noise | $-237.97$ | 0.0 |

The statistical test results, as shown in Table 9 indicate that embeddings generated from noisy data using our GMM-based event extraction approach closely align with embeddings derived from cleaner distributions. Specifically, the two-sample t-test demonstrated no significant difference ($p \geq 0.05$) difference between embeddings from pure data and those obtained from activity data processed with our method. This finding suggests that the GMM-based event extraction effectively isolates meaningful features, ensuring that the embeddings remain robust and consistent with the underlying characteristics of the cleaner dataset. Additionally, we have compared the p-value and t-test with a random noise data, to show how it impacts the statistical measurements.

Table 10: Classification Accuracy (%) of datasets on machine learning methods. Accuracy is in "mean (std.)" format.

| Classifier | Dataset/Events | 2 | 5 | 7 | 10 |
|---|---|---|---|---|---|
| Random Forest | A1 | 81.77(4.52) | 88.10(4.69) | 87.35(4.61) | 90.45(4.25) |
| | A2.1 | 80.86(5.27) | 88.24(4.20) | 90.92(3.46) | 92.75(3.42) |
| | A2.2 | 83.85(5.33) | 89.85(4.45) | 92.17(3.36) | 93.79(3.36) |
| | A2.3 | 85.43(2.65) | 93.55(1.83) | 95.42(1.99) | 97.02(1.47) |
| | A3.1 | 80.41(3.50) | 87.31(3.57) | 89.81(3.44) | 92.46(3.38) |
| | A3.2 | 84.54(2.91) | 91.55(2.67) | 93.98(2.54) | 95.54(2.39) |
| | A3.3 | 80.34(4.54) | 88.06(4.54) | 91.48(3.18) | 93.70(3.35) |
| | A4.1 | 74.63(2.70) | 79.48(3.07) | 80.58(2.73) | 82.45(6.35) |
| SVM | A1 | 74.06(3.26) | 85.52(3.33) | 89.44(3.59) | 92.25(3.77) |
| | A2.1 | 48.74(1.58) | 58.55(2.21) | 61.88(3.21) | 67.04(3.70) |
| | A2.2 | 58.29(3.31) | 70.70(4.99) | 73.22(5.13) | 76.43(5.19) |
| | A2.3 | 56.82(1.71) | 68.74(2.53) | 72.16(2.99) | 75.61(2.54) |
| | A3.1 | 22.06(1.24) | 86.42(3.13) | 89.12(2.78) | 92.26(2.55) |
| | A3.2 | 59.03(1.88) | 87.91(2.62) | 90.01(2.44) | 91.96(2.07) |
| | A3.3 | 81.19(0.25) | 88.08(0.29) | 89.53(0.21) | 91.60(0.23) |
| | A4.1 | 40.37(1.88) | 46.85(1.25) | 44.40(2.67) | 45.49(2.65) |

## 6.5 BASELINE DETAILS AND ADDITIONAL RESULTS

### 6.5.1 MACHINE LEARNING APPROACH

We have conducted experiments with machine learning approach for person identification using two feature sets. The first set contains 134 features described in section 4.1 (main paper), while the second set contains 104 features from (Pan et al., 2017) (see Table 10). We extract additional 104 features for classifying sequences of 2, 5, 7 and 10 footstep events. These features are calculated by averaging the data points across consecutive features in the original dataset. We normalise the feature sets and then implement machine learning algorithms SVM and RF. We have used the same hyper-parameters as in section 4.1 (main paper). A key finding is that the model performs worse on outdoor data compared to indoor data. This is likely because the features in both sets were designed for indoor environments.

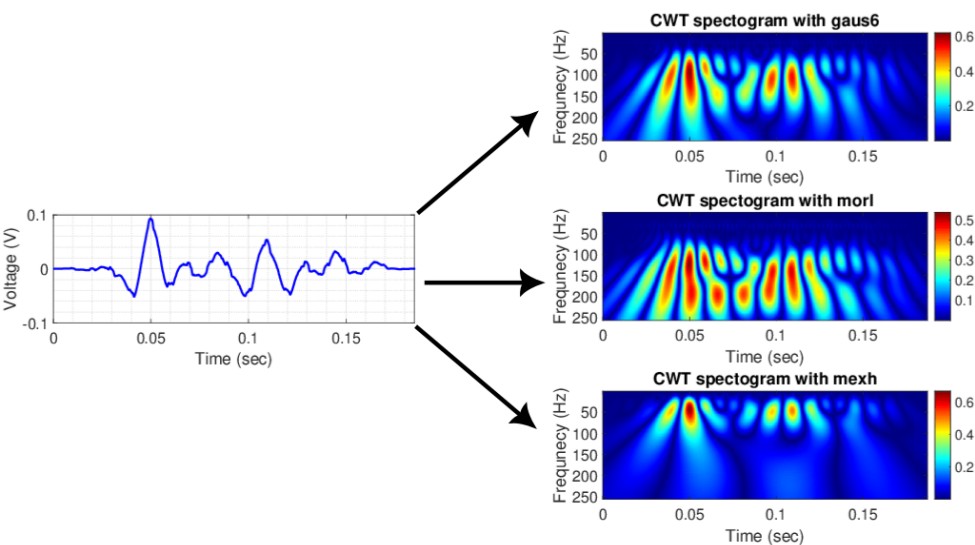

Figure 13: Vibration event and corresponding Continuous Wavelet Transform (CWT) images using different wavelet functions. Data is of P29 participant walking on a cement floor. The CWT with the "morlet" wavelet (morl) function yields the clearest representation of the event.

### 6.5.2 DEEP LEARNING APPROACH

We converted the vibration signal into a time-frequency representation using the Continuous Wavelet Transform (CWT) 0-256 scales. This allows us to analyze the signal's frequency content over time. As shown in the figure 13, we experimented with different wavelet functions to find the best visualization. The " Morlet" wavelet gives the clearest representation of the vibration signal compared to other options.

To evaluate the impact of multi-channel inputs on M models (Table 11), we performed additional experiments using configurations with 2, 7, and 10 events as input, each with an additional Conv2d layer with input channel sizes of 6, 21, and 30, respectively. All input images are in RGB (3-channel) format. While ResNet-18 achieves superior performance in most configurations, this analysis revealed a key trend: M models performs better when large samples are available. Acquiring large datasets and training on it often presents challenges in real-world applications. Therefore, transfer learning can be a promising alternative. To facilitate this approach, we have shared our pre-trained models weights in our GitHub repository. We have used approximately 800 hours of compute power for entire analysis.

Table 11: Classification Accuracy (%) of datasets using deep learning methods. Accuracy is in "mean (std.)" format.

| Classifier | Dataset/Events | 2 | 7 | 10 |
|---|---|---|---|---|
| Resnet-18(M) | A1 | 84.31 (1.54) | 87.35 (3.01) | **93.65** (1.90) |
| | A2.1 | 82.23 (2.40) | 89.23 (2.18) | **91.71** (2.60) |
| | A2.2 | 84.11 (1.31) | 85.74 (2.78) | **89.36** (2.00) |
| | A2.3 | 84.39 (1.74) | 90.70 (2.09) | **91.03** (3.10) |
| | A3.1 | 85.57 (1.06) | 93.70 (2.44) | **94.90** (1.08) |
| | A3.2 | 83.92 (1.51) | 90.77 (2.60) | **92.41** (2.31) |
| | A3.3 | 83.02 (2.54) | 83.10 (2.83) | **88.34** (2.04) |
| | A4.1 | 86.55 (1.24) | **86.92** (3.57) | 80.56 (2.63) |
| Resnet-50(M) | A1 | 82.35 (2.24) | 88.51 (2.61) | **94.70** (2.00) |
| | A2.1 | 81.15 (1.41) | 86.67 (2.12) | **87.71** (2.50) |
| | A2.2 | 81.52 (0.22) | 84.28 (2.80) | **86.17** (2.00) |
| | A2.3 | 83.65 (1.34) | **90.23** (1.71) | 88.37 (2.14) |
| | A3.1 | 81.20 (0.60) | **92.08** (0.81) | 91.07 (1.92) |
| | A3.2 | 82.93 (1.52) | **89.81** (2.10) | 89.72 (2.44) |
| | A3.3 | 81.26 (0.12) | **86.07** (1.50) | 85.95 (1.27) |
| | A4.1 | 81.22 (2.02) | 77.57 (2.71) | **83.33** (3.09) |

### 6.5.3 GAIT ENERGY IMAGE DETAILS

To ensure participant privacy, we have opted to share a curated subset of the video data. This dataset comprises approximately 5,000 normalized human silhouette figures for each 15 participants. Our system extracts individual frames from the video recordings. These frames are then fed into a Mask R-CNN model, to obtain normalised human silhouettes. Sequences of these silhouettes are then directly used to create Gait Energy Images (GEIs). Subsequently, these GEIs are used to train deep learning models for person identification.

### 6.5.4 ANALYSIS ON VARIATION OF SPEED

We conducted an additional sub-experiment with 15 participants (11 males and 4 females) to incorporate different walking speeds: slow (80-90 SPM), normal (90-120 SPM), and fast (120-140 SPM). This experiment yielded 7.5 hours of data, comprising 39,035 total samples. The Baseline results demonstrate that even with variations in walking speed, we achieved good classification accuracy (Table 10 and 11). Participants walked to metronome beats for the slow and fast speeds, while walking at their natural pace for the normal category. Data was collected on a wooden floor, with each participant walking for 10 minutes at each speed, split into two sessions of 5 minutes. The participants' average age is $28 \pm 11.80$ years, height is $1.67 \pm 0.056$ m, and weight is $60.22 \pm 8.63$ kg. Individual details are available on GitHub. We used the same hyperparameters as outlined in Section 4.1 of the main paper. The VIBeID A5 dataset consists of data from 15 participants, walking at different speeds (slow, normal and fast).

- **VIBeID A5.1 :** Data of participants walking at fast speed (120-140 steps per min).

- **VIBeID A5.2 :** Data of participants walking at their normal pace (90-120 steps per min).

- **VIBeID A5.3 :** Data of participants walking at slow speed (80-90 steps per min).

For each comparison, the two-sample t-test was applied to the flattened embeddings to compute the following metrics:

- **T-Statistic:** Measures the difference in means relative to the variation within the embeddings.

- **P-Value:** Indicates whether the observed differences are statistically significant. A p-value $< 0.05$ suggests significant differences.

The results are summarized in Table 12.

Table 12: Pairwise t-Test Results for VIBeID Classes

| Comparison | T-Statistic | P-Value |
|---|---|---|
| A5.2 → A5.1 | 0.9212 | 0.356 |
| A5.2 → A5.3 | -0.069 | 0.944 |

We have trained a ResNet-18 model with normal (A5.2) speed and tested it with the fast(A5.1) and slow(A5.3). The Table 12, shows results indicate that there is no statistically significant difference between the embeddings of the compared classes (p-values (p>0.05)).

Table 13: Classification Accuracy(%) of datasets on machine learning methods. Accuracy is in "mean (std.)" format.

| Feature Extraction Method | Classifier | Dataset/Events | 1 | 2 | 5 | 7 | 10 |
|---|---|---|---|---|---|---|---|
| (Anchal et al., 2020) | Random Forest | A5.1 | 89.95 (2.72) | 93.59 (1.67) | 96.68 (2.51) | 97.75 (1.11) | 97.49(2.05) |
| | | A5.2 | 87.18 (1.54) | 92.74 (0.98) | 96.35 (0.96) | 97.38 (1.13) | 99.01 (0.33) |
| | | A5.3 | 86.22 (1.72) | 91.38 (1.87) | 94.45 (2.19) | 96.33 (1.88) | 97.55 (1.63) |
| | SVM | A5.1 | 49.87 (3.83) | 51.73 (2.17) | 55.85 (3.71) | 60.55 (3.94) | 67.21 (3.47) |
| | | A5.2 | 38.70 (2.01) | 47.95 (1.33) | 56.46 (1.95) | 62.22 (1.54) | 67.73 (1.65) |
| | | A5.3 | 35.79 (1.52) | 39.38 (2.04) | 41.91 (1.88) | 45.48 (1.97) | 49.68 (2.05) |
| (Pan et al., 2017) | Random Forest | A5.1 | 89.16 (1.28) | 93.08 (1.61) | 96.32 (1.52) | 97.25 (1.17) | 97.49 (2.20) |
| | | A5.2 | 86.34 (1.58) | 91.54 (1.12) | 95.20 (1.68) | 95.76 (1.52) | 98.12 (0.86) |
| | | A5.3 | 84.86 (1.54) | 90.30 (1.85) | 94.50 (1.95) | 96.77 (1.58) | 97.46 (1.63) |
| | SVM | A5.1 | 24.27 (1.78) | 35.41 (2.51) | 37.36 (2.61) | 39.29 (2.53) | 39.37 (1.38) |
| | | A5.2 | 20.90 (1.19) | 25.21 (1.00) | 21.54 (1.93) | 18.85 (0.72) | 19.69 (1.89) |
| | | A5.3 | 20.61 (2.23) | 16.09 (4.74) | 16.76 (2.75) | 13.36 (3.62) | 11.76 (4.27) |

Table 14: Classification Accuracy(%) of datasets on deep learning models. Accuracy is in "mean (std.)" format.

| Methods | A5.1 | A5.2 | A5.3 |
|---|---|---|---|
| ResNet-18(S) | **91.69 (0.61)** | **93.49 (0.75)** | **90.62 (1.11)** |
| ResNet-50(S) | 90.39 (0.18) | 91.43 (1.57) | 85.32 (0.15) |
| ResNet-18(M) | 90.06 (1.25) | 89.96 (1.14) | 81.68 (0.34) |
| ResNet-50(M) | 86.55 (1.6) | 85.30 (1.47) | 75.16 (0.68) |

### 6.5.5 QUALITATIVE COMPARISON WITH VISION MODALITY

To evaluate the performance of our geophone-based sensing system relative to vision-based modalities, we designed a series of experiments targeting scenarios where vision systems typically face challenges. The focus is on conditions that compromise the reliability of visual sensing but where geophones exhibit robust and consistent performance. Data was collected under the following scenarios:

- **Normal Conditions**: Environments with stable and adequate lighting, serving as the baseline.
- **Low-Light Conditions**: Scenarios characterized by insufficient illumination.
- **Half Obstructions**: Situations involving physical obstructions that partially occlude the visual field, under normal lighting.
- **Full Obstructions**: Scenarios where physical obstructions completely block the visual field, under normal lighting.

As shown in figure 14, we have used green screen to create obstructions to the line-of-sight of the camera. We performed simultaneous data acquisition for both the geophone and camera systems in these scenarios. Subsequently, feature embeddings were extracted from the collected data and subjected to statistical analysis to compare the performance of each modality under normal and challenging conditions. The analysis utilized an event ratio metric, defined as:

$$\text{Event Ratio} = \frac{\text{Events Extracted under specific Condition}}{\text{Events Extracted under "Normal" condition}}$$

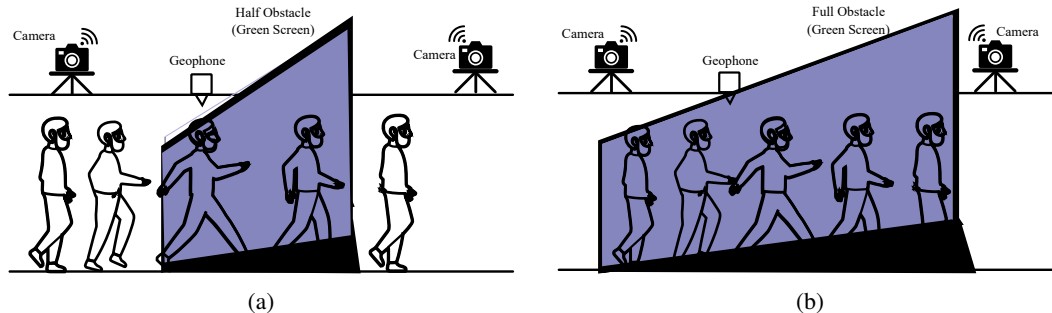

Figure 14: Visual depiction of data collection setup: (a) the camera's field of view is partially obstructed by obstacle, and (b) the camera's field of view is entirely blocked by obstacle.

In this context, an "event" refers to the extraction of meaningful features or representations from the collected data, such as GEI (Gait Energy Image) or spectrograms derived from human silhouettes (in the case of cameras) or geophone signal patterns. These extracted features depend heavily on pre-processing, which varies based on the availability and quality of input data. For embeddings, a pretrained ResNet-18 (IMAGENET1K_V1) (S) was employed by replacing the final fully connected layer with a custom embedding layer of size 512.

For vision-based modalities, the event ratio can be influenced significantly by the quality of the human silhouette extracted from the images. For example:

- Under low-light or partial obstruction conditions, fewer reliable silhouettes may be extracted, leading to a lower event ratio.
- Under full obstruction conditions, where the visual field is completely blocked, no events can be extracted, resulting in an event ratio of zero.

In contrast, geophones are unaffected by visual obstructions or lighting conditions, and their event ratio remains consistent across scenarios. This stability highlights the geophone's robustness in environments where vision-based systems struggle, reinforcing its value as an alternative or complementary modality.

Table 15: Performance comparison of three sensing modalities (Camera 1, Camera 2, and Geophone) across different environmental conditions. Metrics include T-statistics, p-values, and the ratio of events extracted.

| Comparison | Camera 1 | | | Camera 2 | | | Geophone | | |
| Modality | T-Stat. | P-Value | Event Ratio | T-Stat. | P-Value | Event Ratio | T-Stat. | P-Value | Event Ratio |
|---|---|---|---|---|---|---|---|---|---|
| Normal to Normal | 0.0 | 1.0 | 1.00 | 0.0 | 1.0 | 1.00 | 0.0 | 1.0 | 1.00 |
| Normal to Low Light | -4.60 | 0.004 | 0.50 | -2.09 | 0.036 | 0.51 | 0.53 | 0.59 | 1.01 |
| Normal to Half Obstruction | -2.52 | 0.016 | 0.42 | -1.31 | 0.187 | 0.68 | 0.53 | 0.59 | 1.02 |
| Normal to Full Obstruction | - | - | - | - | - | - | 0.461 | 0.644 | 1.00 |

Table 15 provides a comparison of three sensing modalities: Camera 1 (4.2a), Camera 2 (4.2b), and the Geophone (4.1), under varying environmental conditions. Performance is analyzed using T-statistics, p-values, and event extraction ratios. The p-values indicate the statistical significance of differences between data collected under normal and challenging conditions, with values below 0.05 signifying statistically significant performance degradation. Under normal conditions, all modalities achieve consistent performance with an event extraction ratio of 1.00. However, in low-light conditions, Camera 1 exhibits the most significant degradation, with its event ratio decreasing by 50% compared to the baseline, while Camera 2 shows a similar reduction of 49%. Both reductions are statistically significant (p-values below 0.05). The geophone, in contrast, maintains robust performance, with its event ratio increasing by 1%, highlighting its resilience to lighting changes.

In scenarios involving partial or full obstruction, the limitations of vision-based systems become evident. Partial obstruction reduces the event ratios of 4.2a and 4.2b by 58% and 32%, respectively. Meanwhile, the Geophone remains unaffected, demonstrating a slight improvement in its event ratio

(2%). Under full obstruction, visual modalities fail entirely due to the absence of visible human silhouettes, resulting in undefined event ratios. Conversely, the geophone maintains its baseline performance across all tested conditions, underscoring its robustness and reliability in scenarios where visual systems are compromised.

The analysis of p-values further emphasizes the susceptibility of vision-based systems to environmental factors. 4.2a demonstrates statistically significant degradation under low-light ($p = 0.004$) and partial obstruction ($p = 0.016$), while 4.2b shows moderate sensitivity with a p-value of $p = 0.036$ in low-light and no significant differences ($p = 0.187$) under partial obstruction. The geophone consistently achieves high p-values ($p > 0.05$), such as $p = 0.59$ in low-light and partial obstruction, and $p = 0.644$ under full obstruction, indicating stable performance. These results highlight the geophone's potential as a complementary or alternative sensing modality, particularly in challenging environments where vision-based systems fail.

## 6.6 POWER CONSUMPTION AND ENVIRONMENTAL IMPACT

We computed the power consumption of both devices based on their power ratings and usage scenarios, assuming 24/7 operation for one year. The geophone is a passive sensor, meaning it does not require an external power source for operation. The geophone operates within a voltage range of 4.5V to 5.5V, which is compatible with the 5V output provided by the USB port of the Raspberry Pi 3 B+. When powered at 5V, the geophone's current consumption, measured using the Nordic Power Profiler v2 kit, ranges from 5 mA to 10 mA. Therefore, its maximum power consumption by geophone is calculated as:

$$5V \times 10mA = 50mW \text{ or } 0.05W$$

The results are summarized in the table below:

Table 16: Comparison of Power Consumption and Environmental Impact

| Modality | Load | Power (W) | Daily Energy (kWh/day) | Annual Energy (kWh/year) | Equivalent Annual $CO_2$e (kg/year, Global) |
|---|---|---|---|---|---|
| Vision-based | Basic (PoE) | 6.3 | 0.1512 | 55.188 | 26.71 |
| | Maximum (PoE) | 18.9 | 0.4536 | 165.204 | 78.47 |
| Geophone-based | Basic | 1.95 | 0.0468 | 17.082 | 8.11 |
| | Maximum | 5.15 | 0.1236 | 45.114 | 21.43 |

### EQUATIONS AND METHODOLOGY

**Power to Energy Conversion:** Convert Power (W) to Kilowatts (kW):

$$\text{kW} = \frac{W}{1000}$$

**Daily Energy Consumption:** Daily kWh = kW $\times$ hours per day Assuming continuous operation: Daily kWh = kW $\times$ 24 **Annual Energy Consumption:** Annual kWh = Daily kWh $\times$ 365 **Carbon Emissions:** Annual $CO_2$e Emissions (kg/year) = Annual kWh $\times$ Carbon Intensity (kg $CO_2$e/kWh) Global average carbon intensity: 0.475 kg $CO_2$e/kWh (International Energy Agency, 2022).

The operational details of the devices in question highlight their energy consumption profiles and environmental impact. CCTV cameras and Raspberry Pis are designed to operate continuously, 24/7, at specified power levels. For the CCTV camera, the power consumption ranges between a basic load of 6.3W and a maximum load of 18.9W. Similarly, the Raspberry Pi 3B+ operates at a basic load of 1.95W and can go up to a maximum load of 5.15W. To assess the environmental impact of these operations, a global average carbon intensity of 0.475 kg $CO_2$e/kWh is used. It's assumed that the power consumption remains stable during operation, providing a consistent basis for calculating the energy usage and its associated environmental footprint.

As shown in the Table 16, the geophone-based system, with its associated Raspberry Pi, has significantly lower power consumption and carbon emissions compared to the CCTV-based system. This highlights the eco-friendliness of the geophone modality, especially in scenarios requiring continuous operation in an indoor setting. Additionally, a single Raspberry Pi can be modified to record multiple geophone sensors, with very little carbon emission of around 0.208 kg/year per geophone.

### 6.7 LIMITATIONS AND DISCUSSIONS

Structural Vibration-based person identification is an emerging behavioral bio-metric technology. However, we observed it has many limitations, this are as follows:

- **Footwear Dependency:** Our current research focuses on soft-soled flat footwears. Variations in footwear design can alter the induced vibrations. To address this limitation, in the future we will build a comprehensive different footwear dataset. This would enable the model to learn features that are invariant to footwear type, enhancing identification accuracy across diverse footwear choices.

- **Surface Heterogeneity:** Our experiments involved data collection on different floor types and at varying distances from the sensor, which has never been addressed in such details before. We observed a corresponding change in identification accuracy. This highlights the need for further research on domain adaptation techniques within the deep learning community. By leveraging these techniques, the model's ability to generalize across diverse floor surfaces could be significantly improved.

- **Simultaneous person identification:** While our current work focuses on single-person identification, we see substantial potential in extending it to recognize multiple individuals. This would require the development of advanced signal processing and deep learning algorithms inspired by techniques employed in speech recognition and anomaly detection.

Our dataset can serve as a valuable starting point for research in large-scale structural-vibration based person identification. It holds potential for future development, enabling researchers to address current limitations in this field.

### 6.8 POTENTIAL IMPACT TO SOCIETY

Structural-vibration based person identification presents a promising technology for applications in healthcare settings, particularly assisted living communities and nursing homes. These environments prioritize privacy-preserving and non-intrusive monitoring systems. This technology offers a unique advantage by utilizing one-dimensional vibration signals, capturing essential information without requiring intrusive visual data. Our research is ultimately driven by the desire to improve human well-being. By open-sourcing our dataset, we aim to foster greater research interest and accelerate advancements in this field. This approach will contribute to the development of even more robust and effective monitoring systems, ideal for privacy-sensitive applications.

### 6.9 URL TO DATASET

Our project page is hosted at : `https://vibeidiclr.github.io/`. Open-Science Forum: `https://osf.io/4fvnj/`. We have also provided the Human Silhouettes used for experimentation. Each dataset is labelled and annotated. No instance is missing. Complete metadata records for raw data are found in the github page.

