# OpenReview forum: "VIBEID: A STRUCTURAL VIBRATION-BASED SOFT BIOMETRIC DATASET FOR HUMAN GAIT RECOGNITION"
_ICLR.cc/2025/Conference — Submitted to ICLR 2025_

### Official Review · Reviewer_35Gq · 2024-10-29

**Soundness:** 2
**Presentation:** 3
**Contribution:** 3
**Rating:** 5
**Confidence:** 1

**Summary:**

The paper presents a novel dataset termed VIBEID, designed for human gait recognition using structural vibration data. The dataset includes recordings of 100 subjects across various distances, floors, and environments. Experiments demonstrate that structural vibration can serve as a viable biometric trait across different scenarios.

**Strengths:**

- Vibration-based gait recognition introduces a novel approach for human identification.
- The proposed baseline methods are effective.
- This work establishes the largest-scale vibration-based dataset to date.

**Weaknesses:**

- The dataset has a limited number of subjects, although it includes over 88 hours of recorded data.
- Compared to commonly used vision-based gait recognition, the operating distance remains relatively short.
- The evaluation setup lacks clarity.

**Questions:**

1. To what extent do walking speed and the carrying of objects impact recognition performance?
2. Building on question 1, does abnormal gait pose significant challenges for re-identification?
3. The VIBEID dataset studies an operating distance range of 1.5m to 4m, while vision-based gait recognition typically works at distances over 10m. What is the distance limit for vibration sensors to capture meaningful gait signals?
4. If there are obstructions between subjects and the sensor, is reliable recognition still possible?
5. How does the proposed vibration-based gait recognition handle scenarios with multiple pedestrians walking simultaneously?
6. I recommend replacing Figure 3 with a clearer version.
7. The evaluation protocol should be more detailed.

---

> ### Author Response · Authors · 2024-11-20
>
> We sincerely appreciate your thoughtful feedback and suggestions.  Your insights have provided us with crucial perspectives to strengthen our study.
>
> -Dataset size : We would like to highlight that the dataset contains data from 100 subjects. It is important to note that this dataset is first of its kind to address gait recognition using geophone sensors. Additionally, the dataset includes over 88+ hours of recorded data, offering a significant amount of real-world signals that reflect a variety of gait patterns and environmental conditions. We are committed to improve the dataset, by continuously expanding and diversifying the dataset, we aim to enhance its usability for practical applications and ensure compatibility with a wide range of use cases.
>
> -Operating size: The operating range of the geophone is designed to be suitable for environments such as indoor spaces, where monitoring is typically required at close proximity. The distance range can be extended up to 6-10m, by using more better data acquisition units or additional sensors, but this is often constrained by the physical layout of the environment (aka size of the rooms) rather than the inherent limitations of the geophone sensor itself.
>
> -Evaluation setup:  We apologize for any lack of clarity in our evaluation setup. To address your comment, we have rewritten section 4.1 and 4.2.
>
> -1: To investigate the impact of walking speed on recognition performance, we conducted additional experiments, with results presented in Supplementary Section 6.4.4, which includes an extra 7.5 hours of data. Furthermore, given that the data spans over five years, we have recordings from the same individuals in both summer and winter (with heavier clothing), and our observation is that carrying small objects do not significantly affect the underlying gait pattern. However, when individuals carry heavier objects or change their posture (e.g., hunching over), this can alter their gait, potentially leading to abnormal ground contact patterns, which affects recognition performance. Our findings suggest that variations in walking speed and the carrying of objects do indeed affect the gait pattern, as expected. However, the fundamental movement patterns, which our model is designed to recognize, remain consistent despite these changes. We recognize that this is an area that warrants further investigation.
>
> -2: Thank you for your comment. We agree that gait abnormality can pose challenges for re-identification. In our study, we collected data from 100 individuals within a similar age range (25–35), where many of the subjects have similar height-to-weight ratios. This similarity between individuals makes re-identification based solely on standard gait patterns more challenging. However, when a person exhibits a medical condition, this deviation tends to be more distinct and more accessible to detect. The abnormalities in gait patterns are more pronounced and can be identified more accurately. However, anyone deliberately faking abnormal gait would be difficult to detect.
>
> -3: To ensure uniformity in all the scenarios, we limited the range to 4 meters in our experimental design for both indoor and outdoor data collection. This 4 meter limit is not due to any inherent restriction of the sensor's range but instead to the design of the preamplifier circuit and the room sizes used in the study ( Supplementary A-6.1). In practice, our configuration has an effective sensing radius of up to 6 meters indoors and up to 10-15 meters outdoors, depending on the level of background noise.
>
> -4: Thank you for your question. The geophone sensor is capable of recording gait signals even in the presence of obstructions, as it detects ground vibrations rather than relying on line-of-sight. In our experiments, we have successfully recorded data in rooms with various objects, such as tables and chairs, without significant interference to the vibration signals. Therefore, obstructions between the subject and the sensor do not hinder the sensor's ability to capture meaningful gait patterns.\\
>
> -5 : Thank you for raising this important point. our unsupervised event detection module (we have updated in details in supplementary 6.3), can be modified to detect events from multiple activities, by changing the number of clusters from 2 (event vs noise) to 3 (event vs noise vs other activities).
>
> -6 : Thank you for your suggestion. We appreciate your feedback on Figure 3. A revised figure has been included in the updated manuscript.
>
> -7: Thank you for your comment. We apologize for any lack of clarity in our evaluation setup. To address your comment, we have rewritten section 4.1 and 4.2.

---

> > ### Author Response · Authors · 2024-11-25
> > **Follow-Up on Reviewer Feedback**
> >
> > Thank you for your valuable feedback. We have incorporated your comments and hope our edits address them effectively. If you have further suggestions or clarifications, we would greatly appreciate your input to refine our draft before the rebuttal deadline Nov 27 '24 .

---

> > > ### Comment · Reviewer_35Gq · 2024-11-27
> > > **Thanks for your dense response**
> > >
> > > To my knowledge, the primary advantage of gait recognition lies in its ability to identify individuals at a distance. In constrained environments or limited spaces, other commercially available biometric techniques, such as iris or face recognition, may be more practical alternatives to developing gait recognition in such scenarios.
> > >
> > > I acknowledge that visual biometrics like face and iris recognition may raise privacy concerns. To address this, some studies have proposed alternative approaches, such as Lidar-based gait recognition [1], which features a benchmark dataset with over 1,000 subjects under diverse walking conditions. Similarly, non-RGB sensors like event cameras have also been explored for gait recognition [2], with the added benefit of effectively leveraging existing RGB-based gait datasets.
> > >
> > > Both Lidar- and event-based gait recognition methods are privacy-friendly and versatile, suitable for both indoor and outdoor scenarios, across varying distances, and scalable to datasets with thousands of subjects.
> > >
> > > For scenarios involving multiple pedestrians, the proposed method clusters subjects by adjusting the number of clusters. However, manually counting pedestrians is impractical for real-world applications. This challenge may be a unique limitation of vibration-based gait recognition and merits further in-depth discussion in the manuscript.
> > >
> > > While introducing new sensors for gait recognition is commendable and encouraged, top-tier application-oriented publications demand greater clarity and focus on addressing concerns about the method’s motivation and practical viability.
> > >
> > > Thank you for your efforts. However, in its current form, this version of the manuscript may still face challenges in improving my rating.
> > >
> > > References:
> > > [1] LidarGait: Benchmarking 3D Gait Recognition with Point Clouds, CVPR 2023.
> > > [2] Event-Stream Representation for Human Gait Identification Using Deep Neural Networks, T-PAMI 2021.

---

> > > > ### Author Response · Authors · 2024-11-27
> > > > **Thank you for your feedback. Interesting Questions!!!**
> > > >
> > > > We truly appreciate your insightful feedback, and it has helped us rewrite the manuscript with better clarity. Below, we address the points raised and provide clarifications:
> > > >
> > > > **Comparison with Commercially Available Biometric Techniques**
> > > >
> > > > We acknowledge that commercially available biometric techniques may offer an alternate viable solution in constrained indoor environments, each having its own pros and cons. We recognize a deeper, underlying question: why explore structural vibration-based gait recognition when so many established gait recognition methods already exist in the market? Our response is, why not? We believe geophones offer unique potential that is still in its nascent stage and remains unexplored. We envision a future where humans interact effortlessly with diverse sensing modalities, just as naturally as we engage with everyday materials.
> > > >
> > > > In comparison to geophones, LIDAR and DVS sensors rely on maintaining a direct line-of-sight. As discussed in Section 6.5.5., obstructions between the sensor and target can block the line-of-sight (or laser beams), potentially resulting in incomplete or inaccurate data capture. Moreover, geophones present a highly cost-effective alternative to 128-beam LIDAR scanners (1:18 cost comparison on available market products), with significantly lower computational and power requirements (Section 6.6). This makes geophones particularly appealing for applications where affordability and efficiency are critical. We have incorporated a detailed comparison in Table 1 of the revised manuscript to highlight these distinctions. Excerpt shown below:
> > > >
> > > > | Reference                          | Sensor Type     | Subjects | Samples | Domain | Environment |
> > > > |------------------------------------|-----------------|----------|---------|----------|-------------|
> > > > | [Shen et al., 2023](#shen2023lidargait) | Event cameras   | 20       | 4000    | 1        | Indoor      |
> > > > | [Doe et al., 2020](#9337225)      | LIDAR           | 1050     | 25,239 |1| Outdoor     |
> > > >
> > > > **Data Scale and Comparisons**
> > > >
> > > > We sincerely appreciate the reviewer’s point regarding the availability of larger datasets in gait-recognition studies. It is worth noting that early research in these fields also began with relatively small datasets before scaling to thousands of subjects. For instance, even the referenced paper on event-stream recognition includes real-life data from 20 subjects with 4,000 samples. Our dataset provides a detailed exploration of different domains, such as multiple floors, rooms, and sensor distances. Similarly, our current dataset serves as an essential starting point, comparable to foundational benchmarks, which also began with fewer or comparable numbers of subjects [1, 2]. The primary objective of this study is to introduce the most extensive structural vibration-based gait recognition dataset to date, building on prior work (Pan et al., 2017; Mirshekari et al., 2018; Anchal et al., 2020; Dong and Noh, 2023; Xu et al., 2024). This dataset is intended to serve as a foundation for future research, facilitating the expansion and scalability of gait recognition studies to encompass a larger pool of subjects. Additionally, we are committed to ongoing efforts to enhance the quality and scope of the dataset.
> > > >
> > > > **Use of Clustering in Multi-Pedestrian Scenarios**
> > > >
> > > > We wish to clarify that our approach does not depend on manual intervention (or manual calculation of cluster number). Instead, we leverage unsupervised clustering to segment signals into three distinct categories. The intuition behind this is that a signal inherently comprises three types of information: background noise, the person of interest, and all other activities (including human, non-human, and group activities). By consistently setting the cluster number to three, regardless of the number of individuals present, our approach focuses on isolating the person of interest amidst diverse and dynamic activities. Our clustering method is statistically robust, effectively detecting the person of interest even in noisy environments, without the need to propose entirely new methods.
> > > >
> > > > **Baseline and Future Directions**
> > > >
> > > > Our experimental baselines were intentionally kept simple to provide initial insights and to establish a foundation for future research. These methods serve as a starting point for the community to explore more complex approaches. We welcome collaboration and encourage researchers to leverage our dataset for advancing novel techniques in this domain.
> > > >
> > > > We deeply appreciate the reviewer’s insightful comments and ongoing engagement, which have played a crucial role in enhancing the quality of the manuscript.
> > > >
> > > > **References:**
> > > >
> > > > [1] Hiroyuki Yamada et al., Advanced Robotics, 2020.
> > > > [2] Shiqi Yu et al., ICPR, 2006.

---

> > > > > ### Comment · Reviewer_35Gq · 2024-12-03
> > > > >
> > > > > Thank you for your response.
> > > > >
> > > > > 1. There seems to be a possible citation typo in Table 1.  Shen et al. appear to have proposed the LIDAR-based dataset, while Wang et al. are credited with the Event-based one.
> > > > >
> > > > > 2. I am still unclear about how the selection of the person of interest is handled. If five individuals are walking within the field of view of the geophone, how can unsupervised algorithms reliably extract the walking signals of each individual? Moreover, how are these signals matched to specific individuals? I consider this a critical issue for real-world applications. Since this submission primarily focuses on application-oriented contributions, it may be challenging to sidestep this issue by explaining that it is merely a pioneering attempt.
> > > > >
> > > > > 3. I appreciate the value of exploratory work.  In fact, this manuscript serves as a continuation of the efforts outlined in the datasets listed in Table 1.  However, given that it is now 2024, a dataset limited to only 100 subjects may pose challenges for acceptance at a top-tier conference.
> > > > >
> > > > > I agree with the perspectives shared by the other reviewers and appreciate your thoughtful response.
> > > > >
> > > > > However, opinions can vary, and I still feel inclined to maintain my current rating.
> > > > >
> > > > > Thank you for your understanding.

---

> > > > > > ### Author Response · Authors · 2024-12-04
> > > > > > **Thank You for your comments!!**
> > > > > >
> > > > > > Thank you for your valuable feedback.
> > > > > >
> > > > > > - First, we sincerely apologize for the citation typo in Table 1 and have corrected the error.
> > > > > >
> > > > > > - We understand the section needs to be written with more clarity. Our focus is on detecting whether the specific "person of interest" is present in the recording, even when there are other activities happening in the background, such as noise, other humans, or groups. To achieve this, we use a Gaussian Mixture Model (GMM) with 134 features (see Section 3.5). During training, we use data containing only footsteps, with no intentional background activity, and cluster it into two categories: noise (natural structural vibration of the location) and footsteps events (due to impact on the ground) (See Section 3.5).  We understand that real-world scenarios are much more complex. To address this, we conducted an additional experiment using a "wild set" (see Supplementary Section 6.4).  In this scenario, the person of interest walked in the presence of background concurrent activity like non-human, human and group activity. During testing, we clustered the data into three categories. Based on previous research (Anchal et al. 2020), the cluster with the largest covariance determinant is labeled as "Complex Noise"- meaning more variations, while the second-largest is identified as the "Person of Interest" among the three clusters (see Section 6.4).  Instead of separating signals for multiple individuals, our goal is to identify whether the person of interest is present in the recording, even in complex environments. To answer your question about "five individuals," we are not attempting to separate signals for all five people but rather to detect and isolate the signal trace of our specific person of interest. To match the detected cluster containing the person of interest, we perform a T-test on the extracted embeddings from both the actual dataset and the wild set (Supplementary Section 6.4.1). This approach helps us on detecting the signal trace of the "Person of Interest"  amidst complex environments.  We believe this is a starting point for future improvement, and plan to try more complex approaches soon, which will likely require novel methods contributions.  We acknowledge that more sophisticated methods may be needed in the future to fully address scenarios involving multiple individuals with overlapping signals.  Our goal is to provide a simple baseline for future research, and we hope this work inspires further advancements in this field.
> > > > > >
> > > > > > - We understand your concern about the dataset size of 100 subjects for a top-tier conference in 2024. We would also highlight that apart from 100 participant's data, we’ve also explored various use cases, such as different floors, rooms, and sensor distances, along with meta-data including height, weight  for real-life applications. We believe this work is an important first step and a strong starting point, based on previous research. Collecting this data took significant time and effort to ensure its quality and reliability. We hope it will inspire other researchers to build on our work and push the boundaries even further.
> > > > > >
> > > > > > We genuinely hope you see the value of our effort and the potential it holds for advancing this field. Your thoughtful feedback has been immensely helpful, and we deeply appreciate your time and engagement with our work. Thank you for reviewing our submission.

---

> ### Author Response · Authors · 2024-11-28
> **Thank you !!! Thank you again for your valuable feedback.**
>
> Thank you again for your valuable feedback. We hope our edits addressed your comments. We would appreciate any additional feedback, comments or suggestions you might have to further improve our draft before the end of the rebuttal period.

---

### Official Review · Reviewer_LqAV · 2024-11-01

**Soundness:** 3
**Presentation:** 3
**Contribution:** 2
**Rating:** 6
**Confidence:** 5

**Summary:**

The presented work introduces a new biometric dataset for human gait recognition based on structural vibrations. The dataset is applied to various tasks such as person identification, domain adaptation, and multi-modal scenarios combining vibration and vision-based identification methods. Experimental analysis includes verification of machine-learning and deep learning approaches.

**Strengths:**

1. The description of the data collection protocol is clearly written with sufficient details and clear explanations of the research motivation
2. The introduction and related work sections contain important background information justifying the motivation for the introduced dataset.

**Weaknesses:**

1. Is there any requirement for using a specific sensor type during the inference if trained on the presented dataset? I'm wondering about the practical implication of the proposed solution.
2. The work indicates that the concurrent activity was not taken into account, however it's very possible to happen in real-life scenarios. Would the presented dataset be sufficient for handling such scenarios? How should one prepare for additional noise introduced in this way?
3. It's not clear how filtering of potential noise was performed? Was the assumption that the data collection is performed in an isolated environment without any noise? You mentioned that there was environmental noise present, but how do you quantify its presence? If the assumption is that there is minimal or no noise, it again raises question around the practicality of the solution.
4. It's not clear how the data was split for training and testing? Were the same subjects present in both subsets or did you ensure no overlap?
5. One of the motivation behind introducing a new dataset is that other datasets contains a limited number of subjects. It's mentioned that there are 100 subjects in the proposed dataset but then only 30 and 40 subjects are used for floor types and distance measurements. Why not all 100 subjects were used for all of the scenarios?

**Questions:**

1. What do you mean by events in Table 2?
2. Line 407 - where is table 10? or did you mean 1?

---

> ### Author Response · Authors · 2024-11-20
>
> We sincerely appreciate your suggestions.
>
> -  1: Thank you for your question. Currently, our model has been trained and tested exclusively on data from geophone sensors, the core principles and techniques we've developed can be applied to a variety of geophone types. Moreover, it can be used simultaneously along with vision-based systems. Merging the modalities can improve overall efficacy. Additionally, it can be used as an alternative to pressure-mats, or wearables, as it is non-intrusive and does not require direct physical contact.
> Additionally, for the broader machine learning community, it serves as a valuable resource for validating new methods, offering a distinct set of 100 distinct classes, containing structural vibration signals.\\
>
> - 2: We acknowledge the importance of addressing concurrent human activities to enhance the usability of our study. In response, we have proposed a method to isolate footstep events in the presence of concurrent activities, which is detailed in the supplementary section 6.4. Specifically, we have demonstrated how our event detection module can be used to distinguish the person-of-interest from other activities, whether human or non-human. Our structural vibration dataset is highly valuable as it offers labelled data for identifying the person-of-interest. We encourage the community to build upon our dataset and develop innovative solutions for addressing the challenges of gait recognition in real-world settings.
>
> - 3: As highlighted in Supplementary Section 6.3, we have used an unsupervised approach to address the challenge of filtering potential noise from the dataset. Specifically, we use a Gaussian Mixture Model (GMM), which is an unsupervised clustering technique, to differentiate footstep events from background noise. The GMM is effective in identifying the presence of distinct event patterns (footsteps) while grouping other irregular noise patterns into separate clusters, thus isolating the relevant signals from the environmental noise. As GMM is an unsupervised method, it can be applied across different environments without relying on predefined thresholds or assumptions about the type or amount of noise present. This allows for flexible operation in various locations without the need for extensive noise profiling, making it suitable for real-time, adaptable applications.
>
> -  4: You've raised an important point about the distinction between training and testing sets in multi-class classification and human identification. Initially, we used a multi-class classification approach, treating each individual as a separate class. This method, commonly used in previous research (Pan et al., 2017; Mirshekari et al., 2018; Anchal et al., 2020; Dong and Noh, 2023; Xu et al., 2024), allows us to assess the model's ability to differentiate between subjects. To address your concern, we conducted additional experiments (Section 4.2, Tables 4 and 6). These results demonstrate the model's identification accuracy on distinct training and testing sets, where the test set contains unseen individuals, aligning more closely with standard human identification practices.
>
> -  5: The proposed dataset contains data from 100 subjects, not all 100 subjects were used for every scenario due to challenges in maintaining consistent data collection. The data was recorded over a span of five years, and the outbreak of COVID-19 interrupted our data collection process, preventing us from gathering continuous data for all 100 individuals. As a result, for certain sub-tasks such as floor types and distance measurements, we utilized the available subjects (30 and 40 individuals, respectively). We would argue that using 30 and 40 subjects is sufficient for an initial understanding of the effects of different domains on structural vibration signals. Moreover, we believe that the introduction of this dataset, is an important step toward addressing the data scarcity in this field. By open-sourcing the dataset and the code, our goal is to encourage further contributions and foster research in this domain.
>
> -  6: Thank you for your question. In Table 2, the term "events" refers to distinct footstep events, which correspond to impacts made by the human foot on the floor. These impacts are unique and can be distinguished from any background noise, as illustrated in Figure 5.  We use consecutive footstep events for training and testing. For example, "2 events" indicates two consecutive impacts on the floor, corresponding to two steps, while ``5 events'' refers to five consecutive steps. Similarly, "7 events" and "10 events" represent seven and ten steps, respectively. These events are used after pre-processing to train the model, helping to capture the dynamics of human gait.
>
> - 7: Thank you for pointing this out. It appears to be a typographical error, and we meant to refer to Table 2, not Table 10. We have corrected this mistake in the revised manuscript.

---

> > ### Author Response · Authors · 2024-11-25
> > **Follow-Up on Reviewer Feedback**
> >
> > Thank you for your valuable feedback. We have incorporated your comments and hope our edits address them effectively. If you have further suggestions or clarifications, we would greatly appreciate your input to refine our draft before the rebuttal deadline Nov 27 '24 .

---

> > > ### Comment · Reviewer_LqAV · 2024-11-25
> > > **Thank you for rebuttal**
> > >
> > > Thank you for addressing my concerns and performing additional experiments to provide more details on my and other reviewers' comments (low light, concurrent activity, walking speed). I think your work has potential. There are still some valid concerns about practicality of the solution's deployment when it comes to sensor availability/type and dataset size/variety, etc., but with additional improvements added, I'm willing to increase my score by 1.

---

> > > > ### Author Response · Authors · 2024-11-27
> > > > **Thank you !!!**
> > > >
> > > > Thank you for your constructive feedback and for raising your score. We agree that looking into a larger and more diverse dataset and using different types of sensors are great next steps, have emphasized this in the Limitation and Conclusions section. Our dataset is the first of its kind for studying gait recognition with geophone sensors. We are dedicated to continually improving our dataset and making our solution more practical for real-world use. We hope the machine learning community will use our dataset to develop and test new algorithms. Thank you once again for your valuable guidance and support.

---

### Official Review · Reviewer_g5tV · 2024-11-02

**Soundness:** 3
**Presentation:** 3
**Contribution:** 3
**Rating:** 6
**Confidence:** 3

**Summary:**

The authors built a dataset with 100 people using geophone to do multiple experiments based on the human's structural vibration. It consists of multiple covariances including floor types, and distances. The work tries to find a connection between the vibration and identities and builds multiple benchmarks.

**Strengths:**

The paper is clear and easy to follow, and the tables and figures are easy to understand
The proposed question is interesting, trying to build the connection between identity and walking vibration, a fine detail when a human is walking. And authors collect a relatively large dataset in multiple conditions.

**Weaknesses:**

In real-life applications, it is hard to find a good condition to use a geophone to capture a human's gait with little noise.

How to control the noise in outdoor cases.

Compared to a camera, the vibration-based method is restricted by the sensor and distance.

The protocol is not clear. How is the train and test set defined? For human identification, the identities appearing in the training set will not be present in the test set. It seems these experiments do not follow this setting

**Questions:**

Although the authors define the floor in different classes, the hardness might be a more reliable way to classify. Since different carpets' thicknesses may have different responses. And what is the distance range for the geophone, since 4 m is not far for camera sensor

**Details Of Ethics Concerns:**

Involving human data collection

---

> ### Author Response · Authors · 2024-11-20
>
> We sincerely appreciate your thoughtful feedback.
>
> - Comment 1 : In response to your comment, we have conducted additional experiments with concurrent human and non-human activities. Statistical tests (t and p-value) suggest no significant difference between noisy and everyday environments. We acknowledge the challenges of acquiring noise-free signals in real-world scenarios; our research demonstrates the potential of geophones for gait recognition, even in diverse environments. By testing across various flooring types, sensor placements, and outdoor settings, we've shown that geophones can effectively detect and monitor individuals in sparsely populated areas.
>
> - Comment 2 : In outdoor scenarios, directly controlling noise is not possible. Instead, we used an unsupervised event detection module to isolate footstep event from background noise (see Supplementary Section 6.3). While outdoor environments are inherently challenging due to unpredictable noise levels, our methodology is designed to adapt and perform signal extraction under such conditions.
>
> - Comment 3 : We agree that cameras have advantages in specific scenarios; however, exploring alternative modalities like vibration sensing is essential to expand application possibilities. Geophones capture vibrations transmitted through the floor, enabling them to work effectively without requiring a direct line of sight or precise alignment, as cameras do. Additionally, Geophone-based gait recognition does not require physical contact, is immune to lighting conditions, is low-cost, and is less computationally expensive and eco-friendly than using vision-based systems.
>
> - Comment 4:  You've raised an important point regarding the distinction between training and testing sets in human identification scenarios. In our initial experiments, we focused on multi-class classification, where each individual is treated as a separate class. This methodology, commonly adopted in prior research (Pan et al., 2017; Mirshekari et al., 2018; Anchal et al., 2020; Dong and Noh, 2023; Xu et al., 2024), enables us to evaluate the model's capacity to discriminate between different subjects. To address your concern, we have conducted additional experiments, as detailed in Section 4.2, Tables 4 and 6. These results demonstrate the identification accuracy of our model when evaluated on distinct training and testing sets, where the individuals in the test set are not present in the training set. This approach aligns more closely with the typical human identification paradigm.\\
>
> - Comment 5 : The thickness of the carpet used is 9mm, this information is updated in the manuscript. We agree that the thickness of carpets can influence the ground vibrations and, consequently, the geophone's response. To address this, we conducted experiments on three distinct floor types: carpet (soft), wood (medium), and tile (hard). We believe that most real-world carpet variations would fall within the spectrum of these three categories. Our primary goal was to assess the consistency of gait pattern recognition across different surfaces, rather than focusing solely on carpet variations. To further characterize the environmental factors affecting the geophone's signal, we included noise measurements for each room in Supplementary Section 6.1. To ensure uniformity in all the scenarios, we limited our experimental design has a range of 4 meters for both indoor and outdoor data collection.
> This 4-meter limit is not due to any inherent restriction of the sensor’s range but instead to the design of the preamplifier circuit and the room sizes used in the study (Supplementary A-6.1). In practice, our configuration has an effective sensing radius of up to 6-10 meters indoors and up to 10-15 meters outdoors, depending on the level of background noise.
>
> - Ethics Review: Structural vibration signals are one-dimensional signals that capture essential information about human gait, making them inherently more privacy-preserving than other methods. This does not reveal any sensitive details like facial recognition or fingerprints. In our study, we took additional steps to enhance privacy by anonymizing the data at the collection point, ensuring no personally identifiable information could be linked to the vibration data. Sharing this dataset openly is intended to contribute to the field and encourage a broader discussion on the privacy safeguards necessary when using such technologies.

---

> > ### Author Response · Authors · 2024-11-25
> > **Follow-Up on Reviewer Feedback**
> >
> > Thank you for your valuable feedback. We have incorporated your comments and hope our edits address them effectively. If you have further suggestions or clarifications, we would greatly appreciate your input to refine our draft before the rebuttal deadline Nov 27 '24 .

---

> > ### Comment · Reviewer_g5tV · 2024-11-25
> >
> > 1. How to deal with the noise caused by group of people walking together?
> > 2. Vision-based models are also eco-friendly. For instance, the pose-based method (GaitTR) could be smaller than 1M and the silhouette-based (GaitGL) methods are less than 10M, but the ResNet-18 is comparatively larger.
> > 3. Exploring different modalities as input is valuable; however, it is unclear whether the geophones are widely used in daily life or data collection. Consequently, the potential for recognizing gait using this modality appears to be limited.

---

> > > ### Author Response · Authors · 2024-11-25
> > > **Thank you for your feedback.**
> > >
> > > 1. As mentioned in Section 6.4 of the supplementary materials, titled "Wild Set Evaluation Involving Concurrent Human and Non-Human Activity," we have modified our event detection module for detecting both human and non-human activities. Results show that it is possible for detecting person-of-interest from any concurrent activities happening simultaneously. Our current setup with a geophone sensor is really meant for identifying individuals by their walking patterns in an indoor setting. When many people walk together , their steps mix together, making it hard to separate one person's walking pattern from another, and require more in depth analysis. This is similar to trying to understand each person talking at the same time during a phone call, which is possible but needs more detailed study. Our goal with this research is to introduce a large-scale dataset that can serve as a foundation for the machine learning community to further develop and refine. This dataset is intended to inspire and enable more advanced studies in the field.
> > >
> > >
> > > 2. We acknowledge that GaitTR and GaitGL are more efficient in terms of model size compared to ResNet-18. These methods are built on existing datasets for gait recognition and focus on optimizing within those frameworks. In contrast, our work introduces a new dataset specifically designed for novel-modality geophone-based gait recognition. We validated our approach using ResNet models due to their established reputation and widespread acceptance in the machine learning community as benchmark models.
> > > Geophone-based gait recognition captures one-dimensional signals, which provide significant advantages in terms of storage and processing simplicity compared to video-based systems. Video data typically requires substantial storage and computational resources, whereas geophone data is more compact and suitable for resource-efficient preprocessing and real-time deployment on devices with limited storage capacity.
> > > 3. We appreciate this observation and acknowledge that geophones may not yet be widely used in daily life or standard data collection scenarios. While geophones represent a relatively new modality in this domain, previous research has already demonstrated their potential for capturing gait characteristics effectively in Table 1 and section 2 (Pan et al., 2017; Anchal et al., 2020; Dong & Noh,2023; Mirshekari et al., 2018; Chakraborty & Kar, 2023; Xu et al., 2024). Geophones, as highly sensitive vibration sensors, offer a non-invasive, privacy-preserving, and cost-effective alternative to traditional video or image-based systems. Unlike cameras, geophones do not capture visual data, making them suitable for environments where privacy concerns are paramount, such as in healthcare settings. Additionally, geophones can operate effectively in low-light or visually occluded environments, where video-based systems may struggle (supplementary section 6.5.5). Moreover, by introducing a new geophone-based dataset, we aim to pave the way for further research in this area, bridging the gap between emerging technologies and real-world applications. As machine learning and sensor technologies evolve, geophones could find increased relevance in various specialized use cases, ultimately demonstrating their utility for gait recognition and beyond.

---

> > > > ### Author Response · Authors · 2024-11-27
> > > > **A Deep Dive into the Comments !!**
> > > >
> > > > - In response to your comments we explored the possibility of detecting individual persons within a group walking together. Group can be defined two or more person. We considered scenarios where two individuals walk concurrently but independently within the same recording environment.  We recorded an additional 10 minutes of data featuring individuals walking side-by-side in an uncontrolled, random manner. Our goal was to determine whether our modified event detection system could successfully identify specific individuals amidst group activities. We then utilized statistical p-value tests to quantify the impact of such group dynamics on our detection capabilities. (See Supplementary section 6.4)
> > > >
> > > >  ## Statistical Test Results for Pure vs. Activity Data
> > > >
> > > > | Comparison                          | T-Statistic | P-Value |
> > > > |-------------------------------------|-------------|---------|
> > > > | Pure & Non-Human Activity Data      | -0.772      | 0.440   |
> > > > | Pure & Human Activity Data          | -1.750      | 0.080   |
> > > > | Pure & Group Activity Data          | -1.329      | 0.183   |
> > > > | Pure & Random Noise                 | -237.97     | 0.0     |
> > > >
> > > > The statistical test results, as shown in Table 9 indicate that embeddings generated from noisy data using our GMM-based event extraction approach closely align with embeddings derived from cleaner distributions. Additionally, we have compared the p-value and t-test with a random noise data, as a control.
> > > >
> > > > - In response to the question of comparing the eco-friendliness of two modalities, we considered both vision-based and geophone-based systems. While vision-based modalities include models designed with eco-friendly considerations, our analysis emphasizes the act of recording data. This provides a direct comparison of the environmental impact of the two modalities. (See Supplementary section 6.6)
> > > >
> > > > From a power consumption perspective, the geophone is a passive sensor, meaning it does not require an external power source for operation. It generates an electrical signal in response to mechanical vibrations. However, the associated electronics, such as a Raspberry Pi, consume power. Conversely, vision-based systems typically use CCTV cameras, which have more substantial power requirements.
> > > >
> > > > ## Comparison of Power Consumption and Environmental Impact
> > > >
> > > > | Modality        | Load         | Power (W) | Daily Energy (kWh/day) | Annual Energy (kWh/year) | Equivalent Annual CO2e (kg/year, Global) |
> > > > |-----------------|--------------|-----------|------------------------|--------------------------|------------------------------------------|
> > > > | **Vision-based**| Basic (PoE)  | 6.3       | 0.1512                 | 55.188                   | 26.71                                    |
> > > > |                 | Maximum (PoE)| 18.9      | 0.4536                 | 165.204                  | 78.47                                    |
> > > > | **Geophone-based** | Basic    | 1.95      | 0.0468                 | 17.082                   | 8.11                                     |
> > > > |                 | Maximum      | 5.15      | 0.1236                 | 45.114                   | 21.43                                    |
> > > >
> > > > *Note: PoE stands for Power over Ethernet.*
> > > >
> > > > ---
> > > >
> > > > The geophone-based system, with its associated Raspberry Pi, has significantly lower power consumption and carbon emissions compared to the CCTV-based system. This highlights the eco-friendliness of the geophone modality, especially in scenarios requiring continuous operation in an indoor setting. Additionally, a single Raspberry Pi can be modified to record multiple geophone sensors, with very little carbon emission of around 0.208 kg/year per geophone.
> > > >
> > > > ---
> > > > Thank you again for your thoughtful feedback. We hope that the edits we have made reflect that we understood your concerns and have addressed them. We would be very grateful if you could let us know whether or not our edits have addressed your concerns and/or changed your opinion on the quality of the paper before the rebuttal period ends.

---

> > > > > ### Author Response · Authors · 2024-11-28
> > > > > **Thank you !!! Thank you again for your valuable feedback.**
> > > > >
> > > > > Thank you again for your valuable feedback. We hope our edits addressed your comments. We would appreciate any additional feedback, comments or suggestions you might have to further improve our draft before the end of the rebuttal period.

---

> > > > > > ### Comment · Reviewer_g5tV · 2024-12-03
> > > > > >
> > > > > > Thanks for your follow-up experiments and explanation. Although I still have concerns about the range of applications and the effectiveness compared to the vision-based method, introducing a new modality is helpful. I would like to increase the score by 3.

---

### Official Review · Reviewer_Sqmc · 2024-11-07

**Soundness:** 2
**Presentation:** 3
**Contribution:** 2
**Rating:** 6
**Confidence:** 4

**Summary:**

This study introduces a benchmark for gait recognition utilizing a novel structural vibration sensing technique, the geophone. and the new benchmark comprised of in total 100 subjects, collected under indoor or outdoor settings. It invesitgated whether the novel sensing modality can encode identity related information, and what is the limitations or sensitivity of this technique. Although this work addresses an interesting topic, it may not yet provide the technical depth or extensive experimental validation expected for broader applicability. There are also several concerns regarding the experimental settings and presentation clarity.

**Strengths:**

This paper presents a well-justified study, especially it clearly identifies current research gap for person identification. Overall it reads very well.

**Weaknesses:**

* [Sample size] Probably for the gait recognition task, we are more interest in how many subjects collected. The subject size, compared with current large dataset, especially the GaitSet, is still not that comparable.

* [Technical contents] I am concerned that the technical content is somewhat limited, even for a benchmark paper, for ICLR. Please consider adding more experiments and tasks to thoroughly validate the usability of this dataset, such as Re-ID, gait event detection, and generalization across subpopulations... I encourage the authors to refer to established works like GaitSet for inspiration. Gait data is highly complex, influenced by factors such as age, gender, emotion, and health conditions. Reflecting on these factors in your experiments would enhance the depth of the study.

* [Applicability] I am also concerned that this kind of ambient sensor can only be applied indoor or with relatively small distances, which might limit its application, compared to wearable data?

* [Experiment setup] In the experiment settings, I noticed that there were no concurrent human activity when recording the data, this may be another issue that limits the usability of this study. Additionally, will the data be sensitive to the perspective of the sensor, as I know it is quite sensitive for vision based person identification.

* [Gait event] gait event detection, is this be validated in terms of accuracy?

* [Dataset details] Further elaboration on the dataset’s composition and subject split for person identification would be valuable, particularly for readers unfamiliar with this topic.

* [Table clarity] Table 5 is not clearly illustrated, what is the performance comparison between structural vibration and camera? Very limited information is given in both the table and the associated texts. Expanding on this comparison would help readers understand the relative strengths of each technique.

* [Minor - clarification] I assume the subjects of A2, A3, A4 are part of A1, correct? Please clarify this.

* [Minor - citation] When citing a work which actually does not play any role in your sentence, please use (X et al., XXXX), rather than X et al. (XXXX).

**Questions:**

I would appreciate if the authors could address the concerns I raised in the weaknesses section.

---

> ### Author Response · Authors · 2024-11-20
>
> We sincerely appreciate your valuable feedback and insightful suggestions.
> - Sample size: We appreciate the your observation regarding the sample size, especially in comparison to popular vision-based gait recognition datasets which often involve larger participant groups. However, we would like to emphasize that VibeID represents a first-of-its-kind experiment and constitutes the largest dataset for gait recognition using geophone sensors—practically 10 times larger than existing datasets in this domain (Pan et al., 2017; Mirshekari et al., 2018; Anchal et al., 2020; Dong and Noh, 2023; Xu et al., 2024). While vision-based datasets are the result of decades of research in the field, structural vibration as a modality for gait recognition is still in its early stages. Our work aims to bridge this gap and highlights the feasibility of geophone sensors as a promising modality for gait recognition. We believe that our study lays a crucial foundation for future research and underscores the importance of exploring this emerging domain.
>
> - Technical contents: In response, we have included a person identification use case in addition to the multi-class classification task initially proposed (as detailed in Section 4.1).Our primary objective in this study is to establish foundational baselines using traditional signal processing, machine learning, and deep learning methods. These baselines are intended to serve as a starting point for the broader machine learning community to build upon this dataset and evaluate their methods.
>
> - Applicability: We appreciate the concern expressed regarding the limited application of ambient sensors to indoor environments. While wearable sensors offer certain advantages, they can be intrusive and uncomfortable for continuous monitoring. Structural vibration sensors, such as geophones, offer a non-intrusive alternative for gait analysis. They can be easily be installed in various settings, including homes, and hospitals. For instance, in healthcare settings, geophones can be installed in patient rooms to monitor gait patterns remotely, without requiring patients to wear any additional devices. The small distance monitoring limit is not due to any inherent restriction of the sensor's range but instead to the design of the preamplifier circuit and the room sizes used in the study ( Supplementary section 6.1). To fully harness the potential of ambient sensor technology like geophones, it's essential to continue exploring innovative approaches and addressing current limitations like lack of large-scale datasets.
>
> - Experiment setup:  We acknowledge the importance of addressing concurrent human activities to enhance the usability of our study. In response, we have proposed a method to isolate footstep events in the presence of concurrent activities, which is detailed in the supplementary section 6.4. Specifically, we have demonstrated how our event detection module can be used to distinguish the person-of-interest from other activities (human or non-human). To evaluate scenarios involving concurrent human and non-human activities, we conducted additional experiments that yielded an additional 30 minutes of data. A geophone signal is generally less dependent on perspective compared to vision-based identification systems. Unlike vision systems that need specific angles to accurately capture features, geophones detect vibrations transmitted through the floor. This means they can reliably capture gait information without needing a direct line of sight or a precise orientation toward the individual. This characteristic allows geophones to operate effectively across a range of placements, as they pick up on the unique patterns of movement through floor vibrations rather than visual cues.
>
> - Gait event: As highlighted in in Section 3.5, we have used a pre-existing validated toolkit for gait event detection in both structural vibration and Gait Energy Image (GEIs). Both the toolkit has been tested and validated in previous studies (Song et al.2022, Anchal et al. 2020).
>
> - Dataset details : Thank you for your feedback, we have re-written the entire section 4.1, and highlighted the splits. Additionally, we have added Table 2, which provides more detailed information about the dataset's composition and subject split for person identification.
>
> - Table clarity: We have re-written the entire section, and highlighted the Identification accuracy with train-test splits and explained the results in detail. As shown in the Table 7, GEIs are somewhat vulnerable to view-points, whereas structural vibration is not.  By leveraging the complementary strengths of both modalities, we hope to develop more reliable and efficient gait recognition systems in the future.
>
> - Minor -  Yes, A2, A3, and A4 are subsets of A1. We have now clearly stated in the main text to avoid any confusion.
>
> - Minor - citation:  We have corrected the manuscript.

---

> > ### Comment · Reviewer_Sqmc · 2024-11-21
> > **Thanks for the rebuttal**
> >
> > Thanks for the rebuttal. However, I am still concerned about the technical contributions and applicability of this work, especially comparing against vision based work. The authors argued that "Geophone-based gait recognition does not require a direct line-of-sight or physical contact, immune to lighting condition, low-cost, and is less computationally expensive and eco-friendly that using vision-based systems. ", I would expect some experiments and results to demonstrate Geophone can do what vision cannot do -- this would be particularly important.
> >
> > And also a very minor point. I can see the efforts of changing the citation format. Unfortunately current format is still wrong. Although this minor point would not affect my final recommendation regarding this paper at all, I would like to remind that, in the official latex template of ICLR, it already said using \citep{}, rather than manually adding bracket ().

---

> > > ### Author Response · Authors · 2024-11-23
> > > **Thank you for your feedback. Excellent Question !!**
> > >
> > > We understand your concerns regarding the technical contributions and the applicability of this work, particularly in demonstrating the unique strengths of Geophone-based sensing in comparison to vision-based systems. We have addressed this concern with two approaches:
> > >
> > > - Quantitative Analysis :
> > > We have experimented with metrics mean average precision (mAP), for multi-modal datasets (Table 7 ). Quantitative results indicate that the geophone-based modality achieves performance comparable to vision-based modalities. While vision-based systems exhibit lower mAP in certain scenarios, the geophone modality demonstrates consistent performance. However, this could be attributed to the need for view-invariant analysis in vision-based systems . To provide definitive evidence that the geophone can succeed where vision-based systems fail, we extended our study with qualitative analysis to demonstrate its unique advantages in challenging scenarios.
> > >
> > > - Qualitative Analysis :
> > > We  conducted a series of experiments to  highlight scenarios where the geophone outperforms vision-based systems. We recorded videos and structural vibration signals in varied conditions, namely, normal, low-light, half obstructions, and full obstruction. Obstruction was created using a green screen placed in the field of view of the camera, while recording data (Figure 14). We conducted statistical test-like t-test, p-value test. Additionally, we evaluated each modality on events extracted.  These qualitative results, as discussed in detail in Supplementary Section 6.5.5 (Table 15- excerpt shown below).
> > >
> > > | **Comparison**           | **Camera 1 T-Stat.** | **Camera 1 P-Value** | **Camera 1 Event Ratio** | **Camera 2 T-Stat.** | **Camera 2 P-Value** | **Camera 2 Event Ratio** | **Geophone T-Stat.** | **Geophone P-Value** | **Geophone Event Ratio** |
> > > |---------------------------|----------------------|-----------------------|---------------------------|----------------------|-----------------------|---------------------------|-----------------------|----------------------|---------------------------|
> > > | Normal to Normal          | 0.0                 | 1.0                   | 1.00                      | 0.0                 | 1.0                   | 1.00                      | 0.0                   | 1.0                  | 1.00                      |
> > > | Normal to Low Light       | -4.60               | 0.004                 | 0.50                      | -2.09               | 0.036                 | 0.51                      | 0.53                  | 0.59                 | 1.01                      |
> > > | Normal to Half Obstruction| -2.52               | 0.016                 | 0.42                      | -1.31               | 0.187                 | 0.68                      | 0.53                  | 0.59                 | 1.02                      |
> > > | Normal to Full Obstruction| -                   | -                     | -                         | -                   | -                     | -                         | 0.461                 | 0.644                | 1.00                      |
> > >
> > > Under normal conditions, all modalities achieve a consistent event ratio of 1.00. In low-light conditions, Camera 1 and Camera 2 experience significant degradation, with event ratios dropping by 50% and 49%, respectively, and statistically significant p-values (\(p < 0.05\)). In contrast, the geophone remains robust, showing a slight improvement in its event ratio (+1%). Under partial obstruction, the event ratios for Camera 1 and Camera 2 decline by 58% and 32%, respectively, with Camera 1 showing significant degradation (\(p = 0.016\)). The Geophone remains unaffected, with its event ratio improving by 2%. In scenarios of full obstruction, vision-based systems fail completely, producing undefined event ratios, while the Geophone maintains consistent performance.
> > >
> > > The Geophone’s high p-values (\(p > 0.05\)) across conditions, including \(p = 0.59\) in low-light and partial obstruction, highlight its resilience and stability. These results highlight the Geophone's reliability as a complementary or alternative sensing modality, particularly in scenarios where vision systems struggle. Its independence from environmental factors, low computational cost, and privacy-friendly design further enhance its appeal.
> > >
> > > We hope this clarification addresses your concerns and demonstrates the technical contributions and real-world applicability of our work. Thank you for your thoughtful feedback, and we welcome any further questions.

---

> > > > ### Author Response · Authors · 2024-11-25
> > > > **Follow-Up on Revisions**
> > > >
> > > > Thank you again for your valuable feedback. We hope our edits addressed your comments. We would appreciate any additional feedback to further improve our draft before the end of the rebuttal period.

---

### Meta-Review · Area_Chair_c4uP · 2024-12-12

**Metareview:**

The paper introduces VIBeID, a comprehensive dataset for human gait recognition using structural vibration, addressing the limitations of prior small-scale datasets. VIBeID includes over 88 hours of vibration data from 100 individuals, collected across different floor types (wood, carpet, cement) and distances (1.5m, 2.5m, 4.0m) from a geophone sensor, with additional multi-modal data combining vibrations and video recordings. It establishes benchmarks for person identification, domain adaptation, and multi-modal comparison, showcasing the effectiveness of machine learning and deep learning methods, such as ResNet-18 and ResNet-50, for identifying individuals based on their unique walking-induced structural vibrations. The dataset demonstrates structural vibration as a non-invasive, privacy-preserving biometric modality suitable for diverse environments, with significant potential for applications in security, healthcare, and smart buildings. Future work aims to expand the dataset and explore new applications, further solidifying structural vibration's role in soft biometrics.

The paper is well written and the dataset is composed of quite new modality for human gait recognition. However, some concerns are still not addressed:
- Dataset Scale: Recent gait recognition datasets using emerging sensors like Lidar typically involve thousands of subjects, which makes the size of VIBEID appear limited in comparison. The feasibility of geophone-based gait recognition has already been demonstrated by pioneering works. The small scale of dataset may limit the usage in ML community considering that this data modality is new.
- Feasibility in the real world: The current approach still struggles with simultaneously identifying multiple target subjects. This limitation significantly reduces the system’s practical applicability in real-world scenarios. I do believe single-person also has a value yet the audience may not be ML community (more in ubiquitous computing / mobile computing).

Therefore, I think the paper has novelty but may not be ready for an ICLR publication.

**Additional Comments On Reviewer Discussion:**

In the rebuttal, the authors mainly answer the questions regarding to the details of the datasets: real-world applications, data scale, and clarity. Since it is a dataset paper, not many new experiments are required.

After rebuttal, though the average score is 5.75, three reviewers still have concerns on this submission, e.g., the range of applications and the effectiveness compared to the vision-based method, the selection of the person of interest, and the applicability of this work. I think it has not reached the bar of ICLR.

---

### Decision · Program_Chairs · 2025-01-22

Reject